# Biopolymers Produced by Lactic Acid Bacteria: Characterization and Food Application

**DOI:** 10.3390/polym15061539

**Published:** 2023-03-20

**Authors:** Cristina Mihaela Nicolescu, Marius Bumbac, Claudia Lavinia Buruleanu, Elena Corina Popescu, Sorina Geanina Stanescu, Andreea Antonia Georgescu, Siramona Maria Toma

**Affiliations:** 1Institute of Multidisciplinary Research for Science and Technology, Valahia University of Targoviste, 130004 Targoviste, Romania; 2Faculty of Sciences and Arts, Valahia University of Targoviste, 130004 Targoviste, Romania; 3Faculty of Environmental Engineering and Food Science, Valahia University of Targoviste, 130004 Targoviste, Romania; 4Doctoral School of University of Medicine and Pharmacy “Carol Davila” Bucharest, 050474 Bucharest, Romania

**Keywords:** lactic acid bacteria, polylactic acid, polyhydroxyalkanoates, exopolysaccharides, food application, food packaging

## Abstract

Plants, animals, bacteria, and food waste are subjects of intensive research, as they are biological sources for the production of biopolymers. The topic links to global challenges related to the extended life cycle of products, and circular economy objectives. A severe and well-known threat to the environment, the non-biodegradability of plastics obliges different stakeholders to find legislative and technical solutions for producing valuable polymers which are biodegradable and also exhibit better characteristics for packaging products. Microorganisms are recognized nowadays as exciting sources for the production of biopolymers with applications in the food industry, package production, and several other fields. Ubiquitous organisms, lactic acid bacteria (LAB) are well studied for the production of exopolysaccharides (EPS), but much less as producers of polylactic acid (PLA) and polyhydroxyalkanoates (PHAs). Based on their good biodegradability feature, as well as the possibility to be obtained from cheap biomass, PLA and PHAs polymers currently receive increased attention from both research and industry. The present review aims to provide an overview of LAB strains’ characteristics that render them candidates for the biosynthesis of EPS, PLA, and PHAs, respectively. Further, the biopolymers’ features are described in correlation with their application in different food industry fields and for food packaging. Having in view that the production costs of the polymers constitute their major drawback, alternative solutions of biosynthesis in economic terms are discussed.

## 1. Introduction

Biopolymers are generally defined as polymers from natural/biological sources that are either synthesized from biological material or biosynthesized by living organisms such as bacteria, yeasts, and mold species. According to the standard EN 13432, emphasizing the role of the substrates used in their production, biopolymers are polymers produced from renewable resources [1].

Several classification criteria have been used for biopolymers, considering their origin, chemical structure, and properties, and also their industrial applications. Polymers directly extracted from biomasses (e.g., polysaccharides, proteins), polymers typically produced from bio-derived monomer (polylactic acid; PLA), and polymers synthesized by microorganisms (e.g., polyhydroxyalkanoates; PHAs) are of high interest nowadays, in terms of production, potential application, and biodegradability [2].

Besides their role in dealing with plastic pollution through mechanisms of degrading plastic in natural environments, the microorganisms serve as biosynthetic machinery of bioplastics [3]. The eco-friendly and biodegradable bioplastics produced by microorganisms constitute an alternative to hydrocarbon-derived plastics, and thus contribute to reducing the use of these late-mentioned polymers. Bioplastics or so-called ”green plastics” can be acquired by using renewable resources. This name is also given to plastics with ultimate degradability to carbon dioxide and water [3]. Considered innovative biopolymers, PHAs and PLA are the primary drivers of growth in bio-based biodegradable plastics, whose production is estimated to increase from 0.88 million tons in 2017 to 5.33 million tons by 2026 [4].

With market shares of 1.2% and 13.9%, respectively (weight-related percentages), PHAs and PLA residues constitute at the same time the leading variety of bioplastics [3]. If one refers to the PLA only, in the timeframe from 2017 to 2020, its production increased by 81.5% [4].

Exopolisaccharides (EPS) are known in a wide range of structures and are primarily produced as a response of bacterial species to some anthropic processes. Interesting and attractive characteristics for food are exhibited by EPS produced by lactic acid bacteria (LAB): thickness enhancement, mouthfeel modulation, water-holding capacity, health benefits (e.g., prebiotic properties), as well as the capacity to form films for biodegradable packaging [5,6,7].

Lactic acid bacteria, known from ancient times as agents of uncontrolled fermentation leading to food products to satisfy people’s needs in terms of survival, are nowadays intensively exploited in industrial applications, as starters, as biocontrol agents, and more recently as constituents of certain functional foods (i.e., probiotics). Based on their characteristics, and on new innovative technological strategies, LAB are considered promising microorganisms for biorefineries converting waste biomasses into high-value-added products [8].

High quantities of various sorts of packaging materials are demanded by the food industry each year, out of which plastic materials that are non-biodegradable rank in a significant negative position. Otherwise, the packaging sector is pivotal to the resilience of the EU economy [9]. With a view to sustaining this statement, the European Commission published on 30 November 2022 the Revision of the Packaging and Packaging Waste Directive [10].

According to the European Economic and Social Committee (EESC) opinion entitled ‘’Making packaging a safe, affordable and eco-friendly industry“ [11] over the forecasted period 2021–2026 it is expected that the European consumer packaging market will register a compound annual growth rate (CAGR) of 4%.

This review aims to provide an extensive overview of LAB strains and their potential in the biosynthesis of PLA, PHAs, and EPS, respectively. Strategies to integrate bacterial intervention within the entire production chain by modulating the LAB metabolismand improving the synthesis of these valuable polymers are emphasized. Finally, the wide range of applications of PLA, PHAs, and EPS, respectively in the food industry and the development of food packaging materials is described.

## 2. Classification of Biopolymers Produced by Lactic Acid Bacteria

Biopolymers are linear or branched macromolecules made up of repeating units called monomers. Monomeric units are linked together by covalent bonds [12]. Depending on the nature of the repeating unit, biopolymers can be classified into groups like polysaccharides, glycolipids, lipopolysaccharides, proteins, etc. [8,13,14,15].

Biopolymers can be synthesized by plants, animals, and microorganisms [13,16]. Lactic acid bacteria (LAB) are capable of producing biopolymers with very different chemical structures, grouped into exopolysaccharides (EPSs), polyhydroxyalkanoates (PHAs), and polylactic acid (PLA) [8,14,15].

Based on their chemical composition, two groups of EPSs (Figure 1) can be identified: homopolysaccharides (HoPSs) formed from a single type of monosaccharidesand heteropolysaccharides (HePSs) formed from two or more types of monosaccharides [17,18,19,20,21,22].


**Homopolysaccharides (HoPSs)**


HoPSs are long-chain biopolymers consisting of repetitive units of either sugars or sugar derivatives, mainly glucose, fructose, and galactose [15,18,20,22].

HoPSs have been divided according to the position of the carbon involved in the linkage and the type of linkage in four subgroups: α-glucans, β-glucans, β-fructans, and galactans [23]. α-glucans and β-glucans are formed by glucose polymerization, but their structure is varied due to the different types of linkages formed inside [24]. A specific bond depends on a specific enzyme involved in its activation [16,23].

***(A) α-D-glucans*** contain residues of α-D-glucose linked by α-(1→2), α-(1→3), α-(1→4) or α-(1→6) glycosidic bonds, regularly or randomly distributed, linear or branched in positions 3 (more frequently), and 2 or 4 (less frequently), strain-specific [24,25,26]. α-glucans are classified into the following subclasses: dextrans, alternans, reuterans, and mutans. The enzyme involved in the production of α-D-glucans is glucansucrase. It is synthesized by bacteria of the genus *Leuconostoc*, *Streptococcus*, *Lactobacillus* [27], and *Weisella* [22].

*Dextrans* are primarily made up of a linear chain of D-glucose, linked α-(1→6) glycosidic bonds (95%) and linked α-(1→3) glycosidic bonds (5%) [15], with variable degrees of branching at position α-(1→3) and at positions α-(1→2) and α-(1→4) (less frequently) [15,23,26,28,29]. Dextran is water soluble [22]. LAB-producing dextran are *Leuconostoc mesenteroides* subsp. *mesenteroides* [26,30,31], *Leuconostoc mesenteroides* subsp. *dextranicum* [26,31], *Lactobacillus* genera [32], *Leuconostoc mesenteroides NRRL B-512F*, *Weissella cibaria* [15,28,33], the enzyme involved in dextran biosynthesis being dextransucrase [16,25,26,34].

*Mutans* are made up of D-glucose molecules linked by α-(1→3)-glycosidic bonds alternating with α-(1→6)-glycosidic bonds, in the branching points [22,24,35]. Mutan is water-insoluble [17,22]. The mutans-producing enzyme, mutansucrase, is secreted by some strains of *Leuconostoc mesenteroides* [23,27], *Lactobacillus reuteri* [31], and by many strains of the genus *Streptococcus* (*Streptococcus mutans* [15,31,35], *Streptococcus sobrinus* [15,31], and *Streptococcus salivarius* [22].

*Reuterans* are made up of D-glucose molecules linked by α-(1→4) and α-(1→6)-glycosidic bonds [17,23,24,31,36]. Jurášková and collaborators (2022) and Nabot and collaborators mentioned the presence of α-(1→6) bonds in the branching points [22,23]. Reuterans are water-soluble [22]. The enzyme that synthesizes reuteran, reuteransucrase, is produced by *Lactobacillus reuteri* 121 [15,31,33] and *Lactobacillus reuteri* ATCC 55730 [27].

*Alternans* contain, in equal proportions, α-(1→3) and α-(1→6) glycosidic bonds [18,31,35,36], sometimes branching at the α-(1→3) position [23,34]. They have low viscosity and high water solubility [22,34]. Alternans are produced by the enzyme alternansucrase, synthesized by *Leuconostoc mesenteroides* [22,27,31] and *Leuconostoc citreum* [22,23,33].

***(B) β-glucans (curdlan)*** are branched polymers made up of glucose units linked by β-(1→3) glycosidic bonds, with side chain linked β-(l→2) glycosidic [18,22,31,36]. The enzyme involved, 1,3-β-glucan synthetase, is produced by *Lactobacillus brevis*, *Pediococcus claussenii*, *Pediococcus parvulus*, *Oenococcus oeni* [23], *Lactobacillus brevis TMW 1.2112* [37] or *Pediococcus damnosus* [38], and *Lactobacillus suebicus* [22]. β-glucans can also be produced by *Streptococcus* spp., as mentioned by Jurášková and collaborators, and De Vuyst and collaborators [22,26]. Synthesis of β-glucan occurs intracellularly, according to a mechanism of action not fully yet understood [18].

***(C) β-fructans*** are formed by the polymerization of D-fructose molecules. The degree of polymerization and the type of bonds formed vary according to the producing enzyme [39]. Fructans are produced by the genera *Leuconostoc*, *Lactobacillus* [22,31], and *Streptococcus* [33]. There are two basic types of β-fructans: β-fructans inulin-type and β-fructans levan-type. β-fructans inulin-type have β-(2→1) glycosidic bonds and levan-type β-(2→6) glycosidic bonds, with β-(2→1)-linked side chains [15,24,31,36]. Hundschell et al. and Zannini et al. pointed out that the levan may have several side chains linked β-(2→1) [40,41]. Inulin is slightly soluble in water (maximum 10% at room temperature) [41], but levan has a high solubility in water [17].

The enzyme involved in the production of β-fructans inulin-type is inulosucrase, synthesized by *Leuconostoc citreum*, *Lactobacillus johnsonii* [33], *Lactobacillus gasseri*, *Weissella confusa* and *Weissella cibaria* [23], *Streptococcus mutans*, and *Lactobacillus reuteri* [31], and levansucrase is the enzyme involved in the production of fructans levan-type. Levansucrase is secreted by *Streptococcus salivarius* (SS2), *Leuconostoc mesenteroides*, *Lactobacillus johnsonii* NCC533, *Lactobacillus reuteri* [15,33], and *Streptococcus mutans* [33].

***(D) Galactans (polygalactans)*** are somewhat rarer polymers made up of pentameric repeating units of galactose and have been divided into two groups: α-galactans and β-galactans [41]. They are produced by a number of LAB strains: *Weissella confusa* [22], *Lactococcus lactis* subsp. *lactis*, and *Lactobacillus delbrueckii* subsp. *bulgaricus* [22,36], and are water-soluble.

*α-galactans* have a backbone consisting of a galactose chain linked by α-(1→6) and α-(1→3) glycosidic bonds [23]. Kavitake et al. have identified, using nuclear magnetic resonance (NMR) spectroscopy, a linear polymer with α-(1→6) glycosidic bonds produced by *Weissella confusa* KR780676 [42].

*β-galactans* contain galactose units linked either β-(1→3) or β-(1→6). β-galactans are produced by LAB belonging to the genera of *Lactobacillus*, *Streptococcus*, and *Leuconostoc* [24].


**Heteropolysaccharides (HePSs)**


HePSs contain repeating units of different monosaccharides that range from trisaccharides to octosaccharides [18,19,21,24], with linear or branched chains (at positions C2, C3, C4, or C6) [41]. The monosaccharides can be present in α- pyranose/β- pyranose or α- furanose/β- furanose [43].

## 3. Biopolymers-Producing Lactic Acid Bacteria Strains

Lactic acid bacteria (LAB) have been empirically used, since ancient times, as starter cultures for the production of fermented foods and beverages and for preservation [8,44,45]. Due to their long history of safe use in human consumption [18], some LAB strains received the status Qualified Presumption of Safety (QPS) by the European Food and Safety Authority (EFSA) [46] or Generally Recognized as Safe (GRAS) by Food and Drug Administration (FDA) [47].

LAB comprise a heterogeneous group of genera [45,48] including *Lactobacillus*, *Lactococcus*, *Leuconostoc*, *Pediococcus*, *Streptococcus*, *Enterococcus*, and *Weissella*, known for their wide industrial applications. Other representatives of LAB belong to *Aerococcus*, *Alloiococcus*, *Carnobacterium*, *Dolosigranulum*, *Oenococcus*, *Tetragenococcus*, and *Vagococcus* genera [49].

Members of the genera *Lactobacillus*, *Leuconostoc*, *Pediococcus*, *Lactococcus*, and *Oenococcus* are considered GRAS [15,22,24,26,45]. LAB that belong to the genera *Streptococcus* and *Enterococcus* contain some opportunistic pathogens [50], and are not eligible for GRAS status. Safety concerns arising from their virulence factors and resistance to a variety of antibiotics [45] are associated with members of the genus *Enterococcus*; thus, they were not proposed for QPS status [51].

Important physiological properties are characteristics of all LAB, such as the capacity to ferment carbohydrates primarily into lactic acid via homo- or heterofermentative metabolism [45,50] and the inability to synthesize porphyrin groups (e.g., heme).

LAB are Gram-positive, tolerant anaerobic, catalase-negative, cytochrome-deprived, non-spore-forming bacteria, with rod or coccus shape and with high tolerance at low pH [45,49,52]. Bacilli or cocci may appear as single or grouped cells, in tetrads and short or long chains [53]. These morphological characteristics emphasize the heterogeneity of the LAB group [45]. Lactic acid bacteria are intrinsically resistant to many antibiotics [50].

LAB are generally associated with nutritionally rich environments, because they are nutritionally demanding, with high requirements sources of carbon and nitrogen [53]. The optimum growth for LAB occurs at pH 5.5–5.8 [49], but they can also survive at pHs of around 5 and lower [8]. LAB are commonly found in vegetables, dairy and meat products, beverages, soil, and sewage, as well as in the gastrointestinal and genital tract of humans and higher animals [45,54].

Starter cultures of LAB with industrially important functionalities were developed in the last two decades, offering several technological, marketing, and health advantages, in order to meet the requirements of both producers and consumers. In industrial processes, LAB prove adaptation to stress conditions [8], such as acidic environment, temperature, salt concentration, etc. The optimal growth temperature, depending on the LAB genus and strain, ranges between 20 °C and 45 °C.

Following certain metabolic pathways, LAB produce organic acids (mainly lactic acid, but also acetic acid), ethanol, antibacterial compounds (bacteriocins, hydrogen peroxide), vitamins, enzymes, aroma compounds, EPSs, etc. [18,55]. Citrate utilization results in diacetyl, acetoin, and 2,3-butanediol, whereas amino acid catabolism leads to volatile compounds and bioactive peptides [45]. Depending on the metabolites’ profile, LAB are used in different industrial applications. Production of bacteriocins, bioactive peptides, and antifungal compounds by some LAB is exploited to extend shelf life and enhance microbial safety of food, whereas compounds such as organic acids, volatile compounds, and exopolysaccharides contribute to the sensory and textural profile of some end-products [44,45,50]. Further, the LAB metabolic features allow for maintaining or even enhancement of the nutritional value of someraw materials [45]. Recently, LAB were used for the probiotic features of some strains, based on their ability to colonize the gastrointestinal tracts and proven competitiveness against pathogenic bacteria [50]. A constant increase in the market of functional foods was observed in the last years, with probiotics occupying an important segment, extending from dairy products to a wide range of non-dairy food products (such as vegetable-based, cereal-based, and sweet products).

### 3.1. Lactic Acid Bacteria (LAB) Taxonomy

LAB were classified based on cellular morphology, mode of glucose fermentation and sugar utilization patterns, and also their range of growth temperature [49]. The first classification of LAB was designed by Orla-Jensen in 1919 [45]. Since then, the creation of new genera and species has led to significant changes in the taxonomy of LAB. According to Von Wright [56], LAB belong, in current taxonomy, to the phylum *Firmicutes*, the class *Bacillus*, the order *Lactobacillae* and the families *Aerococcaceae*, *Carnobacteriaceae*, *Enterobacteriaceae*, *Lactobacillaceae*, *Leuconostocaceae*, and *Streptococcaceae* [45]. Figure 2 shows the LAB classification, with a particular emphasis on the strains known as biopolymers producers.

In presenting the LAB taxonomy, the reclassification of the genus *Lactobacillus* as it was proposed by Zheng et al. [57] was taken into account.

Often considered true LAB, the genus *Bifidobacterium* is phylogenetically unrelated to these ones [45]. It belongs to the phylum *Actinibacteria*, with numerous species being nowadays used in industrial processes as probiotics.

Of huge interest for food applications, including the production of biopolymers, the genera of the families *Lactobacillaceae*, *Leuconostocaceae*, and *Streptococcaceae* are mentioned in the paragraphs below. Thus, the family *Lactobacillaceae* includes the genera *Lactobacillus* and *Pediococcus*, the family *Leuconostocaceae* includes the genera *Leuconostoc*, *Weissella*, *Fructobacillus*, and *Oenococcus*, and the family *Streptococcacea* includes the genera *Lactococcus*, *Streptococcus*, and *Lactovum* [58]. A merger of the families *Lactobacillaceae* and *Leuconostocaceae* has been proposed recently [57].

### 3.2. Main Biopolymer-Producing LAB

This part aims to provide an extensive overview of the main genera of LAB involved in the biosynthesis of PLA, PHAs, and EPS, respectively. Metabolism pathways and LAB nutritional requirements are emphasized in correlation with LABs’ ability to synthesize the above-mentioned biopolymers with application in the food industry.


**
*Lactobacillus genus*
**


Lactobacilli are Gram-positive bacteria, non-spore-forming, bacilli or coccobacilli, and aerotolerant or anaerobic [45]. Most strains are mesophiles. Members of the genus *Lactobacillus* are most commonly given GRAS status [50]. A total of 261 different species belonging to the genus *Lactobacillus*, with a relatively high degree of diversity [45], were reported in March 2020 [57]. A reclassification of the genus *Lactobacillus* into 25 genera, out of which 23 are new, was proposed, and it is based on genome phylogeny, whole genome sequences, and ecological and metabolic properties of bacteria [57]. Among the newly proposed genera, are mentioned: *Lapidilactobacillus*, *Lacticaseibacillus*, *Latilactobacillus*, *Lactiplantibacillus*, *Limosilactobacillus*, *Fructilactobacillus*, *Acetilactobacillus*, and *Lentilactobacillus*. The generic term ‘’lactobacilli’’ refers to all organisms classified as *Lactobacillaceae* [45].

Lactobacilli can be found in diverse habitats offering available carbohydrates, such as food, plants, and wastewater, and also in the oral and gastrointestinal tracts of humans and animals.

Lactobacilli are highly exigent from a nutritional point of view, and, besides carbohydrates, they need nitrogen sources (amino acids, peptides), fatty acids esters, vitamins, etc. Members of the genus *Lactobacillus* are strictly fermentative. Accordingly, once glucose is the carbon source, lactobacilli can be homofermentative, producing more than 85% lactic acid, or heterofermentative, producing lactic acid, acetic acid, ethanol, and carbon dioxide [59]. Based on their fermentation characteristics, lactobacilli are classified as follows:

(i) Group I: obligate homofermentative lactobacilli, including the species *L. acidophilus* and *L. delbrueckii.* The members of this group ferment almost exclusively (>85%) hexoses via the Embden–Meyerhof–Parnas pathway (EMP). These organisms are unable to ferment pentoses and gluconate, due to the lack of phosphoketolase.

(ii) Group II: facultative heterofermentative lactobacilli, including the species *L. plantarum*, *L. casei*, and *L. paracasei.* Hexoses are fermented to lactic acid almost exclusively via the EMP pathway. These organisms ferment not only hexoses but also pentoses because they possess both aldolase and phosphoketolase.

(iii) Group III: obligate heterofermentative lactobacilli, including the species *L. sanfranciscensis* and *L. rossiae*. The members of this group ferment carbohydrates in a heterofermentative manner, because they have phosphoketolase but not aldolase. Hexoses are fermented via the phosphogluconate pathway and pentoses can also be fermented if they enter this pathway.

L-lactic acid producers were reported as follows: *L. rhamnosus* growing on starchy biomass and on cassava powder, *L. paracasei* growing on glucose, on rice straw hydrolysate, as well as on food waste. In batch fermentation, *L. platarum* produced L-lactic acid using raw corn starch as substrate. In fed-batch fermentation, *L. delbrueckii* utilized molasses, corn steep liquor, and soybean meal, respectively, and produced D-lactic acid [60]. Six strains of *L. delbrueckii* ssp. *bulgaricus* were found to produce D-lactic acid from orange peel waste, with a yield ranging from 84% to 95% [61].

Lactobacilli have a wide range of applications in the food industry, in both production and preservation. Many representatives of this group are well recognized for their capacity to acidify the environment, to improve the taste and the nutritional value of foods, and relatively recently to act as probiotics, conferring beneficial health effects to the host [45]. From the total L-lactic acid production at large-scale, 90% is attributed to *Lactobacillus* strains and *Bacillus* strains [60]. Most lactobacilli are mesophiles. Using thermotolerant strains (such as *L. rhamnosus* that can grow up to 42 °C) may minimize contamination problems during lactic acid production [61].

D-lactic acid is less studied because it is not metabolized by the human body,. However, the demand for D-lactic acid is currently increasing due to increased requests for PLA. Thus, two wild-type strains, *L. delbrueckii* and *L. bulgaricus*, respectively, were reported to be homofermentative D-lactic acid producers [61]. Moreover, engineered *L. lactis* was carried out to produce a high yield of D-lactic acid from lactose or whey-derived lactose.

Once *Lactobacillus* species as efficient producers of lactic acid was discussed, the same cannot be stated regarding the EPS biosynthesis by lactobacilli compared to other LAB [5]. Mainly HePS are produced by *Lactobacillus* species, with a decrease in their concentration at the end of the fermentation being reported. 

Among *Lactobacillus* EPS-producing species, *L. plantarum* is well investigated in terms of the various functions exhibited by the polysaccharides synthesized. The HePS produced by different strains contained glucose, galactose, fructose, mannose, and arabinose. It was reported that glucose used as carbon source and yeast extract in growth media, respectively, and an incubation time of 72 h improved the EPS production by *L. plantarum* [5].

EPS production by *L. fermentum* was also studied, being emphasized that, as in the case of *L. plantarum*, the medium composition (specifically the nitrogen proportions and type of carbon source) and culture conditions (namely the pH) lead to increased EPS yields [5].


**
*Pediococcus genus*
**


The genus *Pediococcus* comprises Gram-positive, obligate homofermentative, and anaerobic to microaerophilic bacteria. Pediococci microscopically appear coccoidal or ovoid and are commonly found in single cells, pairs, tetrads, or clusters.

*Pediococcus* species involved in winemaking worldwide are known as nutritionally fastidious bacteria, and are generally undesirable because of their ability to produce off-odors and flavor [62]. Currently, there are 11 species of pediococci recognized, with various strains being isolated from different environments, including fermented meat and fermented vegetables.

Considering D-lactic acid as an important monomer of the biodegradable PLA plastic and applying metabolic engineering strategies in *Pediococcusacidilactici*, Qiuand collaborators developed a strain that showed accelerated D-lactic acid fermentation from the lignocellulosic substrate [63]. Thus, a higher yield of D-lactic acid was synthesized by integrating the gene of the short-chain dehydrogenase encoded by *Corynebacterium glutamicum* in the D-lactic acid bacterium strain.

EPS production by *Pediococcus* species involved in winemaking is worrying. These bacteria can synthesize β-glucans composed of D-glucose, which affect negatively the wine’s palatability in concentrations above 100 mg/L [62]. Some strains of pediococci are implicated in causing ‘’ropiness’’ during wine aging. However, although evidence is lacking (including the conditions that favor EPS synthesis), literature quoted by Wade suggests that EPS production by *Pediococcus* sp. may benefit biofilm formation [62].


**
*Leuconostoc genus*
**


The *Leuconostoc* genus includes obligate heterofermentative, mesophilic (with an optimal growing temperature of 30 °C), and acidophilic bacteria [64].

The members of this genus are Gram-positive cocci (single, in pairs, or in short chains), with irregular morphology. *Leuconostoc* species are facultative anaerobes and require complex growth factors and amino acids. They are heterofermentative and use the phosphoketolase pathway for producing D-lactate, ethanol, CO_2_, and small amounts of acetate from glucose. Moreover, some species can convert citrate to diacetyl and acetoin. *L. mesenteroides* also produce mannitol [5]. There are *Leuconostoc* strains able to synthesize dextrans from sucrose [5,55], *L. mesenteroides* being able to produce dextran up to 20 g/l (a feature successfully exploited in the food industry). The dominant *Leuconostocs* in milk and fermented dairy products are *Ln. lactis* and *Ln. mesenteroides* subsp. *mesenteroides* [45]. Production of EPS by *L. mesenteroides/pseudomesenteroides* isolated from fruit and vegetables, including those traditionally fermented in Romania, was evaluated at 15 g/L [5]. *L. lactis* isolated from avocado was also identified as an EPS producer, with a yield of 2.25 g/L.

Numerous applications in the food industry were reported (i.e., in the dairy industry even if they display a low growth rate compared with other starters).

*L. mesenteroides* is a GRAS (Generally Recognized as Safe) organism.


**
*Weisella genus*
**


*Weisella* genus belongs to the *Leuconostocaceae* family and contains species that are gram-positive, catalase-negative, and facultatively anaerobic. Cells are either short rods or ovoid. They are hetero-fermentative, producing lactic acid, acetic acid, ethanol, and CO_2_.

*Weisella* spp. were found frequently in spontaneously fermented foods, such as cocoa beans, fermented sausages, fermented fruits and vegetables, cheese, sour milk, African cassava, sourdough, etc. [65,66,67], participating in their characteristics. However, the species of this genus are not used as starters, being not approved for commercial use in the European Union nor in the United States, although it is assumed that they can enhance the safety, nutritional, and sensory characteristics of food [66]. Thus, *Weisella* species produce bacteriocins that can be used as bio-preservatives, have probiotic potential, show resistance to low pH and bile salts, and can increase the bioavailability of polymeric phenolic compounds. Besides these benefits, some species (*W. cibaria* and *W. confusa*) were identified as high producers of exo-polysaccharides (dextran up to 36 g/L by *W. cibaria*) which exhibit particular features, mainly texturizing properties [5,66], but also can be used as prebiotics in the food industry [65]. Besides dextran, some *W. cibaria/confusa* strains were identified as fructan producers.

Growing evidence related to EPS-producing *Weisella* was obtained by Malang et al., (2015), who identified three distinct phenotypic groups by evaluating 123 strains of *W. confusa* and *W. cibaria* isolated from spontaneous African cassava and sour milk [67]. The authors identified the strains producing (i) dextran only, (ii) dextran and fructan of either levan or inulin-type, and (iii) strains producing dextran and a capsular polysaccharide, respectively. It was emphasized that *Weisella* strains producing more than one EPS type are of special interest in food applications due to their potential synergistic effects on texture and nutritional improvement.

All the above-mentioned characteristics are opening new perspectives in the use of *Weissella* species as starters, although first of all they must be recognized as GRAS.


**
*Lactococcus genus*
**


As well as the members of the genera *Lactobacillus*, the members of *Lactococcus* genera are most commonly given generally-recognized-as-safe (GRAS) status [50]. The genera *Lactococcus* consists of 17 species, including *Lc. lactis* (subspecies *cremoris*, *lactis*, *hordniae*, and *tructae*) and *Lc. plantarum.* These organisms are Gram-positive cocci that occur singly, in pairs, or in a chain. They are facultatively anaerobic, catalase-negative, and resistant to bacteriophages. The glycolytic pathway is characteristic of lactococci, with the predominant end-product of glucose fermentation being L-lactic acid.

From the technological perspective, two characteristics are of significant importance: these organisms can grow at 10 °C and at 40 °C, but not at 45 °C, and can also tolerate high concentrations of NaCl.

Considering their metabolic stability, lactococci are mainly used as starter cultures for obtaining dairy products, such as various cheeses and butter. Many of the traits which render lactococci suitable for fermentations are encoded on plasmids. Traits such as exopolysaccharide (EPS) production have been associated with extra-chromosomal plasmid DNA [55].


**
*Streptoococcus genus*
**


Streptococci are characterized as Gram-positive cocci typically arranged in chains or pairs [48]. More representatives of this genus are facultative anaerobes, but some strains require carbon dioxide for growth. They have complex nutritional requirements and produce lactic acid and other organic acids by carbohydrate fermentation.

Streptococci are found in the commensal microbiota of humans and animals. Some strains, such as *S. pneumonia* and *S. pyogenes*, are pathogenic to humans. Differentiating from other streptococci, *S. salivarius* subsp. *thermophilus* is widely used as a starter culture in the dairy industry, due to its several biochemical properties, including the production of exopolysaccharides.


**
*Enterococcus genus*
**


The members of this genus are Gram-positive cocci that occur singly, in pairs, or in short chains. Enterococci are facultative anaerobes and catalase-negative. They are homofermentative and produce L-lactic acid from glucose fermentation via the glycolytic pathway. Enterococci are heat-tolerant, salt-tolerant, and can grow at a pH of 9.6.

Although several strains are useful for technological applications, the enterococci are also used as indicators of fecal contamination of foods, and some strains were identified as potential pathogens.

Selected studies on biopolymers produced by lactic acid bacteria in different environments are presented in Table 1.

Classification of the polymers produced by LAB and discussed below arementioned in Figure 3.

## 4. Polyesters from LAB

### 4.1. Polylactic Acid (PLA) Production Associated with LAB

Polylactic acid holds a leading position within the group of bio-degradable and bio-based plastics if rigid applications are discussed. However, different modification methods are applied to improve its performance in terms of heat stability and water barrier properties [80].

The polymerization process of LA into PLA, conducted since 1932, has been reviewed thoroughly in the literature. Two much-known methods of PLA production, namely direct poly-condensation (DPC) of lactic acid and ring-opening polymerization (ROP) are reconsidered nowadays aiming to eliminate the disadvantages of the chemical transformation of LA into PLA.

From a chemical point of view, polylactic acid (PLA) is a polyester synthesized via lactic acid (LA). L or D isomers of lactic acid are produced through microbial fermentation of starch-rich agricultural products and then these monomers are chemically polymerized to obtain PLA. The monomers can be polymerized into pure poly-L-LA (PLLA), pure poly-D-LA (PDLA), or poly-D-LLA [81]. The physical properties of PLA are in a relationship with its enantiomer content [3]. Moreover, the morphological and mechanical characteristics of PLA are determined by the presence of different amounts of L-LA and D-LA monomers or oligomers [60].

Homopolymers of PLA are semicrystalline, whereas PLA heteropolymers are amorphous. Homofermentative methods are preferred because they lead to a higher yield of lactic acid with fewer by-products. This method uses *Lactobacillus* sp. such as *Lactobacillus bulgaricus*, *L. delbrueckii*, and *L. leichmannii* [82].

The three stages of PLA synthesis are well known, consisting of LA production (1), LA purification followed by cyclic lactides formation (2), and polycondensation of LA or ring-opening polymerization (ROP) of the cyclic lactides (3) [81]. Both polycondensation and the ROP method exhibit disadvantages. Thus, although the polycondensation is less expensive, it does not give a solvent-free high-molecular-weight PLA. The ROP route involves complicated and expensive purification steps and uses heavy metals as catalysts, as their residues are incompatible with applications of PLA for food contact surfaces [81]. Consequently, attention was focused on replacing the heavy metals catalysts with safe and environmentally acceptable alternatives and overcoming the challenge of the complete biosynthesis of PLA [83].

The biosynthesis of lactic acid is described in the following paragraphs, as the first stage of PLA production. Further, very recent efforts oriented towards designing the entire PLA production a bioprocess by developing alternatives to the DPC and ROP methods, namely, establishing a whole-cell biosynthetic system with recombinant microorganisms, are detailed.

#### 4.1.1. Lactic Acid Biosynthesis for PLA Production

Lactic acid (2-hydroxy propionic acid), a widely occurring carboxylic acid in nature, was discovered by the Swedish chemist Scheele in 1780 [80]. It is considered one of the most important building-block chemicals in the world [61] due to its wide range of industrial applications.

Two different isomers, namely L(+) or S lactic acid (the *dextrorotatory* form) and D(-) or R lactic acid (the *levorotatory* form) are produced by lactate dehydrogenases present in living organisms. Naturally formed lactic acid is usually in L-form. 

Lactic acid (LA) can be produced through chemical synthesis and microbial fermentation from carbohydrates [84] (Figure 4). Only the lactic acid manufactured by the microbial fermentation process is appropriate for the production of poly (lactic acid) (PLA), because the racemic mixture of D-LA and L-LA obtained as a result of chemical synthesis is not desirable for the food industry due to the metabolic problems that D-LA may cause [80]. An important observation to be pointed out is that if the chemical routes lead to the production of a racemic form of D-/L-lactic acid, the microbial fermentation leads to optical pure D- or L-lactic acid [61]. The species of microorganisms responsible for lactic acid fermentation and the specificity of its lactate dehydrogenase (LDH) converting pyruvic acid to lactic acid determine the type of isomers produced. Information on the biosynthesis of D-LA is relatively scarce compared with information on L-isomers [60].

The LAB strains use homofermentative or heterofermentative metabolic pathway(s) to catabolize the sugars. There are also strains that show a mixed acid fermentation phenotype [8].

The homofermentative way occurs according to the Embden–Meyerhof pathway where lactic acid is the only acid produced, and in high amounts. LA production shows higher yields once the homofermentative microorganisms were coupled [84]. The carbohydrates are broken down to pyruvate viathe glycolytic pathway. Further, the nicotinamide-adenine-dinucleotide (NAD)-dependent L-lactate dehydrogenase (EC 1.1.1.27) and NAD-dependent D-lactate dehydrogenase (EC 1.1.1.28) are the enzymes converting pyruvate into L-lactic acid and D-lactic acid, respectively.

The heterofermentative way specific to some LAB following the phosphogluconate and phosphoketolase pathway leads to different products, and lactic acid is one of them [61], together with ethanol, acetic acid, and CO_2_.

A fast-growing and high-yield microbial strain with low-cost nutrient requirements supports the competitiveness of the industrial processes of lactic acid production. The costs related to typical anaerobic fermentation are due especially to carbohydrates needed in broth medium. Optically pure lactic acid production and reducing costs were acquired recently with the help of genetically engineered *E. coli* strains [80].

Recovering the lactic acid from the broth after fermentation, and then its purification, represents challenges of LA production. Thus, lactic acid is adsorbed in suitable polymeric adsorbents [85]. If strong alkali (i.e.,calcium hydroxide) adsorbent is added to the fermentation broth, lactic acid is converted to its basic salt and then desorbed by adding strong acid (i.e., sulfuric acid). Alternatively, reactive extraction can be used to isolate lactic acid from the fermentation medium. Over time, strategies for the clean production of lactic acid were searched. This way, the traditional addition of calcium carbonate as a neutralizer of LA was replaced with the utilization of sodium hydroxide which makes the operation environmentally friendly [60].

Alkalophilic microorganisms growing optimally at a pH above 9 and tolerant to salt were proposed for increased LA production and also for the development of the separation coupling fermentation process. No lactic acid bacteria were reported as alkaliphilic strains.

Clean production of lactic acid can be obtained by using membrane-based hybrid reactor systems. Lactic acid purification is made by a two-step electrodialysis system. Briefly, bacteria and proteins are removed from the fermentation broth by microfiltration membrane, a process followed by nanofiltration. The clarified fermentation broth is concentrated by electrodialysis and then transformed into lactic acid [60].

#### 4.1.2. PLA Biosynthesis

PLA is often not biosynthesized [1], and no enzyme that specifically catalyzes LA to produce lactyl-CoA in nature is known [60]. A whole-cell biosynthetic system, consisting of a one-step synthesis of LA-based polyesters by developing a lactate-polymerizing enzyme was reported for the first time by Taguchi in 2008 [83]. This enzyme substituted the heavy metal catalysts [83] and the microbial process that replaced the chemo-process of LA-based polyesters lead also to the advantage of removing the requirements regarding extremely pure monomers and high temperatures [81]. The authors established a recombinant *Escherichia coli* generating lactyl-CoA as a substrate that can be polymerized by the PHA synthase engineered. Enlargement of the diversity in a combination of monomers available for polymer production and biosynthesis of PLA from renewable carbon sources were identified as attractive and further extensions of the microbial factory developed by Taguchi [83]. 

The enzymatic polymerization of LA monomers for PLA production was recently defined as one of the most viable and environmentally friendly alternative methods [85]. Furthermore, the PLA produced directly by fermentation exhibits high strength and improved yields as compared to conventional methods, important features from a technical and economical point of view. However, developing propionyl-CoA transferase and PHA synthase, the two key enzymes needed for the direct one-step fermentative process of PLA production, is the most critical issue [60].

Recently, an exogenous D-lactate dehydrogenase gene from *Lactobacillus acetotolerans* HT was introduced into various *E. coli* strains [86]. An enhanced LA fraction in P(LA-co-3HB) based on the first-time reported strategy of using ldhD gene expression was considered by the authors preferable not only for effective LA-based polymer production but also for cell growth cultivated on glucose under microaerobic conditions.

PLA biosynthesis started, as it was mentioned above, by its production in recombinant *E. coli*. Due to the significantly low content of PLA produced, not economically for industrial scale, subsequent studies focused on providing more precursors LA by engineering metabolic pathways of host strains and increasing the yield of PLA [60,86]. A future direction should also take into account improving the substrate specificity of PHA synthase [60].

### 4.2. Polyhydroxyalkanoates (PHAs) Production by LAB

Polyhydroxyalkanoates (PHAs) represent a group of high-molecular-weight (about 105 Da) [64] biopolyesters, namely, polyhydroxybutyrate (PHB), polyhroxyvalerate (PHV), and derived polymers viz, poly(3-hydroxybutyrate-co-3-hydroxyvalerate) (PHBV) [3] that are entirely degradable. The monomers of PHAs are always in the R(-) configuration due to the stereo-specificity of PHA synthases [64]. Accordingly, the PHAs exhibit several features similar to oil-derived plastics.

PHAs are the only plastics exclusively produced by microorganisms [82], more specifically by bacterial anabolism [64]. PHAs are synthesized by bacteria as a stress response to the lack of essential inorganic nutrients (i.e., deprivation of nitrogen and phosphorus) and also in the situation of their growth phase [87]. Agro-industrial byproducts, e.g., milk and cheese whey can be subjected to microbial fermentation to obtain PHAs [64].

Diverse microorganisms produce and store PHAs as sources of carbon and adenosine-triphosphate (ATP). Species of *Pseudomonas*, *Alcaligenes*, and *Bacillus* are PHA-producing microbes. In more than 90 genera of microbial species documented, more than 150 different monomer constituents contain straight, branched, saturated, unsaturated, and aromatic structures in PHA [81].

*Leuconostoc mesenteroides* [62], *Lactobacillus plantarum* [14], and *Lactobacillus bulgaricus* [68] are PHAs producing lactic acid bacteria (Table 1). *Lactococcus*, *Lactobacillus*, *Pediococcus*, and *Streptococcus* genera growing on MRS broth were reported as poly-β-hydroxybutyrate (PHB) producers [8], although the obtained yields were lower than the ones obtained in soil bacteria [68]. Recently, the mixed microbial cultures (MMCs), including LAB, growing on food wastes and other suitable biomasses are widely used for PHAs synthesis.

In the last years, decreasing of PHAs production costs by developing alternative processes to pure culture fermentation processes was the focus of research works. Thus, two alternative processes were developed, namely the use of low-cost substrates coming from agro-industrial waste streams and that of MMCs [88] consisting of diverse bacterial genera. Engineering the microbial consortium by using the ecological selection principles was named recently eco-biotechnology. Activated sludge wastewaters, molasses, vegetable oil effluents, wheat and rice bran, and cheese wheyare only a few examples of substrates used to produce PHAs from MMC. A favorable impact on both PHA production and waste disposable costs could be reached by using waste materials as carbon sources for microbial-derived PHA production [89]. However, choosing the most suitable substrate is challenging, because the microorganisms’ metabolism and nutritional requirements must be carefully taken into account for high-yield PHA production. Moreover, the pre-treatments of the candidate carbon source and the choice of the PHA-producing strain are still hindered issues [89]. Thus, although milk whey is one of the most promising carbon-rich substrates, good PHA producers have displayed poor growth on lactose, whereas only a small part of microbial metabolism is directed to PHA production by good lactose utilizers [89].

The MMC PHA production requires lower operating costs because it does not need growth-medium sterilization prior to fermentation. Besides this advantage, MMCs are able to adapt to industrial waste complex substrates. Culture selection is the key to the effectiveness of MMC PHA production processes. Despite the above-mentioned advantages of using MMCs for PHAs production, more recently it was emphasized that better metabolic performances can be reached by using pure cultures of efficient PHA producers [64].

Better metabolic performances on whey with respect to PHA production, yet poorly explored, were attributed to pure cultures of lactic acid bacteria, evolving in the milk ecological niche [64]. From this starting point, the authors isolated from an MMC grown on dairy byproducts (cheese and scotta whey) PHA-producing strains, finding *L. mesenteroides* as one of the most active PHA-producing bacterial populations.

Co-culture fermentation systems including LAB and *Cupriavidus necator* known for their ability to produce PHAs have also been reported [8]. Briefly, the lactic acid produced by LAB by conversion of carbohydrates is taken up by *C. necator* to producePHAs. Although recent literature is scarce concerning the development of co-culture fermentations for PHAs production, numerous related advantages are estimated to sustain future application of co-cultures, such as increased yield with improved control of product qualities and the possibility of utilizing secondary products, cheaper than glucose [8].

## 5. Exopolysaccharides from LAB

Exopolysaccharides (EPSs) are polymeric carbohydrate molecules, namely extracellular polysaccharides that are either associated with the cell surface as capsules, called capsular exopolysaccharides (capsular EPSs), or secreted into the extracellular environment as slime, called slime exopolysaccharides (slime EPSs) [18,19,26,90].

EPSs’ role is to store energy and protect the bacterial cell against unfavorable environmental factors such as temperature, pH, osmotic pressure, desiccation, light, phagocytosis, bacteriocins, protozoa, and toxic compounds (toxic metal ions, sulfur dioxide, ethanol, and antibiotics) [24,26,91]. EPS production and secretion start during bacterial growth and stop in the stationary phase [26,92]. They are synthesized intracellularly and secreted outside the cell, or are produced extracellularly by enzymes secreted by lactic acid bacteria [22].

Bacterial EPSs have a wide range of industrial applications (i.e., food, medicine, pharmaceuticals, and cosmetics) depending on their physicochemical and structural properties. The costs of production are related to the costs of the carbon sources and the EPS yield. Thus, the bacterial EPSs entering the market are relatively limited. The first microbial EPS that was commercialized was dextran [93]. 

Identifying new bacteria producing EPSs at high yields and also with functional features led recently to much interest in lactic acid bacteria [94]. EPS yield is strain-specific and heavily influenced by the substrate used in terms of the nutritional and growing conditions. Food wastes (FWs) are seen as an excellent choice for EPSs production by LAB, both to minimize environmental contamination and also to generate economically relevant EPSs [94]. 

Many review articles discussed the production of the exopolysaccharide by synthetic LAB strains and their physical, chemical, and biological properties related to specific applications in the food industry and health [95,96]. Most EPS-producing LAB belong to the genera *Lactobacillus*, *Streptococcus*, *Lactococcus*, *Leuconostoc*, and *Weissella*. Approximately 30 species of lactobacilli have been reported to produce EPSs, especially *L. casei*, *L. acidophilus*, *L. brevis*, *L. curvatus*, *L. delbrueckii* subsp. *bulgaricus*, *L. helveticus*, *L.rhamnosus*, *L. plantarum*, *L. johnsonii*, etc. [97,98].

LAB may synthesize EPSs (heteropolysaccharides or homopolysaccharides) within an enormous structural diversity [99,100,101] that are differentiated by their monosaccharides’ composition, molecular mass, size, and structure [18,26]. Some possible physiological roles of EPSs are to help LAB in their survival [102] and to offer LAB protection from stress conditions (such as environmental pH, osmotic stress, lack of essential elements such as nitrogen, protection from bacteriophages, antibiotics, lysozymes, etc.) [52,100,103]. EPSs production by LAB occurs not only under growth-limiting conditions but also in the presence of excess available carbohydrates (i.e., sucrose) [103]. The formation of mucoid colonies in solid media and the increase in viscosity in liquid media is the basis of the detection of the presence of EPSs associated with bacterial cells [103]. EPSs are loosely attached to the cell or secreted to the environment [67].

EPSproduction is strain-dependent and is strongly affected by the processing conditions (i.e., carbon source and nutrients existing in the culture medium, pH and temperature, incubation time, etc. [18,95]).

The monomer blocks are polymerized at the cell wall, and EPSs are either liberated into the medium (free EPSs) or remain attached to the bacteria (capsular EPSs). Some LAB strains produce both forms, others only free EPSs. According to the ropy character of the fermented milk, the free EPSs can be further classified [69].

Mentioned distinctively from EPSs [67], the capsular polysaccharides (CPS) are covalently bound to the cell surfaceand structurally can be of the HoPS or HePS type. According to these authors, EPS and capsular polysaccharide LAB producers are frequently belonging to the genera *Lactobacillus*, *Leuconostoc*, *Streptococcus*, *Lactococccus*, and *Weissella*. The greatest variety of structures was reported in lactobacilli [99].

EPSs exhibit a broad range of physic-chemical functionalities and applications [52,95]. Thus, microbial EPS are recognized as bio thickeners due to their stabilizing, emulsifying, viscosifying, or gelling capacity [26] and particularly contribute to the sensory and rheological properties of fermented foods, as well as to their stability [52]. Further, EPS confer unique properties to fermented food, properties that are generally beneficial to humans [102,103], which is why some of them fulfill the criteria considered for functional foods. Recently, EPS were considered functional postbiotic ingredients in fermented foods [52], due to their human health benefits, such as immuno-modulation, anti-oxidative, anti-inflammatory, anti-microbial, or microbiome modulators. The EPS production by probiotic LAB seemed to be responsible for their health effects, such as LAB persistence in the gut ecosystem [104].

However, despite the above-mentioned benefits of EPS produced by LAB, with the exception of homopolysaccharide dextran, until now, only the in situ application of EPS-producing LAB (i.e., as starter cultures) has been economically viable [52]. This is due to the low yield of EPS production by LAB (in comparison with other EPS-producing strains), the required steps of EPS purification, as well as production costs. It was suggested that yields should be in the range of 10–15 g/L for an economically feasible production of EPS to use as a food ingredient [24].

### 5.1. Homopolysaccharides (HoPS)

Depending on the linkage type and the position of the carbon involved in the bond, the HoPS produced by LAB, which display high molecular masses (up to 10^8^ Da) [52], are mentioned in Table 2.

HoPS are mainly biosynthesized extracellularly from an existing molecule of sucrose [18], by the action of glycosyl hydrolase (GH). Thus, glucansucrases (GH family 70) (www.cazy.org) synthesize α-glucans which has been reported for the following LAB genera: *Lactobacillus*, *Leuconostoc*, *Streptococcus*, *Pediococcus*, and *Weissella* [18,100], *Lactiplantibacillus*, *Limosilactobacillus* and *Lacticaseibacillus* [100]. β-fructans are formed by fructansucrases (GH family 68). 

Synthesis of β-glucan occurs intracellularly by a glucosyltransferase (GTF), according to a mechanism of action not fully yet understood [18]. The process does not use sucrose as a substrate [100]. The production of β-glucan by the membrane-associated GTF was described in LAB (i.e., *Lactobacillus*, *Pediococcus*, and *Oenococcus*) isolated from alcoholic fermented beverages, such as cider and wine. Moreover, GTF is part of the mechanism responsible for HePS synthesis [26].

If the production of EPS by LAB is discussed in the light of their potential utilization, after extraction and purification, it should be noted that the yields of HoPS produced by LAB (Table 1) are low as compared with other bacterial EPS [18]. *L. mesenteroides* is the most commonly used bacteria for the industrial production of dextran [93].

### 5.2. Heteropolysaccharides (HePS)

HePS are composed of a backbone of repeating subunits that consist of three to eight monosaccharides, derivatives of monosaccharides, or substituted monosaccharides [18]. These complex polysaccharides, with a molecular weight varying between 10^4^ and 10^6^ Da, are constituted mainly from glucose, galactose, and rhamnose, whereas other monosaccharides (ribose, mannose, fucose), N-acetylated monosaccharides (N-acetyl-D-glucosamine and N-acetyl-D-galactosamine) might be present with lower frequency [18,26]. Moreover, organic (glucuronic acid, glycerol, etc.) and inorganic (phosphate, etc.) substituents can be found in some HePS. The constituents D-glucose, D-galactose, and L-rhamnose are usually joined by β-(1,4) or β-(1,3) and α-(1,2) or α-(1,6) linkages [52]. 

It was reported that *Lactiplantibacillus plantarum* strains are able to produce HePS containing glucose and galactose, arabinose, mannose, glucose, and galactose, and arabinose, rhamnose, fucose, xylose, mannose, fructose, galactose, and glucose, respectively, with all of them having health benefits and applications in the food industry [100]. *Lactobacillus gasseri* produce a HePS composed of glucose, mannose, galactose, rhamnose, and a small fraction of fucose used as viscosifying agent and antimicrobial agent [100]. Strains belonging to *Streptococcus thermophilus* and *Lacticaseibacillus rhamnosus* (formerly *Lactobacillus rhamnosus*) were also identified as HePS producers, but in small amounts [52]. Zeidan reported recently that of 81 structures (out of which 47 were synthesized by *Lactobacillus*, 28 by *Streptococcus*, and 6 by *Lactococcus*, respectively) elucidated so far, 55 are unique [99]. Gellan, xanthan, and kefiran are examples of HePS [45].

The best known is kefiran, composed of mannose, glucose, and galactose in a ratio of approximately 1:5:7 [27]. Zannini et al. mentioned almost equal proportions of glucose and galactose in kefiran produced by *Lactobacillus kefiranofaciens* subsp. *kefiranofaciens*, isolated from kefir grains. Kefiran is a water-soluble polymer [41].

The capacity to produce kefiran has been reported for *Lactobacillus kefiranofaciens* [7], *Lb. kefir*, *Lb. parakefiri*, *Lb. kefirgranum*, *Lb. delbrüeckii* subsp. *bulgaricus*, and *Lb. plantarum* [27].

The structural composition of the EPS produced by *Lactococcus lactis* revealed the existence of β-1,3-linked glucose, β-1,4-linked glucose, and terminal β-galactose [72]. In the HSQC (heteronuclear single quantum coherence) spectrum of EPS produced by *Streptococcus thermophilus*, the signals were assigned as α-1,6-linked galactose, β-1,4-linked glucose, and β-1,4-linked galactose, respectively.

The formation of HePS requires sugar nucleotide intermediates as precursors [67]. The HePS repeating units are intracellularly synthesized, and polymerized outside the cell [18]. Exocellular polysaccharides synthesized by LAB are mostly branched [99]. The LAB growth and their central carbon metabolism influence the HePS biosynthesize [67]. Moreover, this process is influenced by medium composition and its sugar composition, temperature, pH, vitamins, and minerals [100]. The HePS production by LAB occurs mainly on the Wzy-dependent pathway [100] and is generally lower than the synthesis of HoPS by LAB [18].

The chemical diversity of the repeating units corresponding to EPS synthesized by LAB has been described by NMR spectroscopy [18,21], the enzymatic machinery, and the corresponding genes encoding enzymes, involved in the synthesis of HePS, being much more complex than those of HoPS.

The EPSs’ physicochemical properties are affected by their composition, molecular weight, spatial arrangements, and ability to interact with proteins [18].

### 5.3. Screening of EPS

Isolation and quantification of EPSs production by LAB require methods that allow for comparing yieldsaccurately. Leroy and de Vuyst [105] reviewed recently the methods for the production, isolation, purification, and quantification of EPSs, offering a detailed description of these ones.

The EPS-producing strains are typically identified on media supplemented with sucrose, by the appearance of slimy/ropy colonies on solid media or viscous solutions on liquid ones [18,77]. On the basis of ultrafiltration and gel permeation chromatography, der Meulen developed a rapid screening method for the EPS produced by LAB strains isolated from dairy and cereal products [106]. This method was coupled with the screening through polymerase chain reaction (PCR) performed with primer pairs targeting different genes involved in EPS production.

Isolation of pure exopolysaccharides, avoiding their contamination with components from the microbial culture medium, represents the first step of the structural analysis and yields of any EPS. According to Torino et al., the protocols of EPS isolation include cell removal, polymer precipitation from the cell-free supernatant, and dialysis and drying of the precipitated polymer [18]. If needed, a new reprecipitation and dialysis step is applied. For cell removal, centrifugation or filtration is used, whereas polymer precipitation is usually made by cold ethanol or acetone addition. Additional purification steps, such as membranefiltration, anionexchange, and/or gel permeation chromatography can be applied to characterize the EPS structure [18] and to explore its potential biological applications [52].

EPS concentration can be determined by the phenol–sulfuric acid method [107] which estimates the neutral carbohydrate content by means of high-performance size exclusion chromatography coupled with refractive index (RI) detection (HPSEC-RI) [18]. The molecular masses and chemical structures are correlated with the applications of EPS in terms of technological characteristics and biological functions [52].

The molecular weight of EPS is measured based on the retention time of the polysaccharide eluted by HPSEC-RI. The techniques used for molecular weight determination represent major improvements in EPS characterization [18]. Moreover, gel permeation chromatography and asymmetric field-flow fractionation were reported asmethods for the determination of molecular weight [52].

Total acid hydrolysis followed by monomer detection can be used for the determination of EPS monomer composition. High-performance anion exchange chromatography (HPAEC) with pulsed amperometric detection can be applied for monomer detection [18]. Furthermore, high-resolution NMR spectroscopy provides additional information (such as the type of constituent monosaccharides, anomeric configuration, the position of glycosidic linkages, etc.) about the structural features of the EPS [18,65]. Liquid-gas chromatography followed by mass spectrometry is widely applied for the determination of the composition of monosaccharides of EPS [52]. Scanning electron microscopy (SEM) or atomic force microscopy (AFM) served for the observation of the microstructures of the EPS [14]. References regarding the evaluation of EPS by one or another of the methods above mentioned can be found in Table 1.

### 5.4. The Influence of the Cultivation Conditions of Lactic Acid Bacteria on the Biosynthesis of Exopolysaccharides

The influence of the sucrose concentration on the yield of EPS during the fermentation of the barley-malt-derived wort by *W. cibaria* MG1 was emphasized by [108]. The yield of EPS was also dependent on the growth conditions. In sucrose-supplemented MRS broth, the amount of EPS (dextran, 5 × 10^6^–4 × 10^7^ Da) produced was (36.4 ± 0.6) g/L, whereas in 10% sucrose-supplemented wort only (14.4 ± 1.2) g EPS/L was reached. The authors emphasized that efficient EPS production by *W. cibaria* MG1 in wort is strongly related to sucrose supplementation. Having in view the complex composition of wort, the positive correlation between EPS production and increasing ratio of sucrose to maltose was underlined.

Homopolysaccharide production from sucrose by different *Leuconostoc lactis*, *Lc. mesenteroides*, *Lc. pseudomesenteroides*, *Weissella cibaria*, and *W. confusa* isolates was tested [109]. MRS sucrose agar was used as a specific substrate and the experiments were performed at 25 °C, 30 °C, and 37 °C, respectively. All these isolates were able to produce EPS from sucrose at 30 °C but not at 37 °C [109]. Several studies reported that the ability of LAB to produce EPS from sucrose is due to the action of glucansucrase or fructansucrase. The glucansucrase sequence length (pb) ranged from 118 to 613 pb for *Lc. mesenteroides*, from 657 to 916 pb for *W. cibaria*, respectively, and from 29 to 234 pb for *W. confusa.*

It was reported recently that calcium ions have significant effects on the biosynthesis of EPS in *Lactobacillus plantarum* K25 [110]. The addition of CaCl_2_ at 20 mg/L in a semi-defined medium (SDM) increased the EPS yield by 40%, until 238.6 mg/L in 24 h of incubation, and also changed the microstructure of the polymer. Interestingly, the growth of *L. plantarum* was not affected by CaCl_2_ addition. A steady production of EPS was registered at 20 to 80 mg/L CaCl_2_, inhibited by increasing the concentration of CaCl_2_ to 100 mg/L.

MRS broth was supplemented with 10 mM CaCl_2_ in order to evaluate the EPS biosynthesis by *Lactobacillus rhamnosus* ZY [111]. Although the EPS production by *L. rhamnosus* was defined as relatively low, it was observed that the addition of CaCl_2_ triggered EPS overproduction. The yields of crude EPS produced by *L. rhamnosus* ZY at 37 °C in the presence of CaCl_2_ decreased from 2203.5 ± 53.2 mg/L after 12 h of fermentation to 273.2 ± 13.6 mg/L after 72 h. A 9.5-fold increase in EPS yield, until 2498 mg/L at 12 h of fermentation, over the yield of cultures grown under anaerobic conditions, was observed by combining treatment with CaCl_2_ and H_2_O_2_. This approach was reported as not successful when applied to *Weissella cibaria* 27 [112], being supposed that the genera *Lactobacillus* and *Weissella* exhibit quite different mechanisms of EPS production. By sustaining this statement, MRS broth was supplemented with 2% (*w*/*w*) sucrose. A significant increase in EPS production by *W. cibaria*, from 82 mg/L to 9838 mg/L, was achieved. Contrarily, the effects of sucrose on EPS produced by different *Lactobacillus* strains at 30 °C showed the same levels, at around 105 mg/L to 253 mg/L. Supplementation of MRS broth with 2% (*w*/*w*) sucrose determined the increase in the yields of EPS in *L. plantarum* 23 from 72 mg/L to 105 mg/L, respectively, and from 199 mg/L to 253 mg/L in *Lactobacillus acidophilus*. Taguchi’s orthogonal array was also used by Yu et al. [112] for optimizing the culture conditions in relationship with EPS extraction. Referring to sucrose’s effect on the production of EPS by *W. cibaria* 27, it was observed that the ratio between the culture with 20 g/L and 60 g/L sucrose is 1.87-fold.

Fermentation temperature and fermentation time significantly affected the level of in situ EPS production by *Lactobacillus plantarum* and *Leuconostoc mesenteroides* strains in Turkish-style fermented sausage (sucuk) samples [113]. In sucuk samples produced by using *Lactobacillus plantarum* 162 R strain the EPS content ranged from 4.68 mg/kg dry matter (fermentation temperature by 14 °C and 8 days of ripening) to 9.97 mg/kg dry matter (fermentation temperature by 18 °C and increment of ripening period till 16 days). If the sucuk samples were produced by using *Leuconostoc mesenteroides* N6, higher values of EPS were determined, ranging between10.02 mg/kg dry matter (14 °C and 8 days of fermentation) and 18.96 mg/kg dry matter (18 °C and 16 days of ripening). The fermentation temperature and strain-specific conditions were considered the most important parameters for EPS production in sucuk manufacturing. The alterations in EPS production levels among strains were explained both in terms of the relationship with extrinsic factors (i.e., fermentation temperature) and intrinsic factors (i.e., genetic mechanism). 

Two strains of LAB, namely *Lactobacillus plantarum* NTMI05 and *Lactobacillus plantarum* NTMI20, were selected from 27 strains isolated and identified from different milk sources, on the basis of their ability to produce EPS [114]. Optimization of the media (MRS broth) composition was carried out by using central composite design (CCD) and response surface methodology (RSM). Referring to the carbon sources tested (optimized at 20 g/L), glucose, followed by lactose, galactose, and sucrose, caused the highest EPS production. It was stated that with “housekeeping enzymes”, LAB use these sugars to produce EPS [115]. If different sources of organic nitrogen (25 g/L) were tested, the EPS production in *L. plantarum* NTMI05 increased from 0.22 g/L (tryptone) to 0.37 g/L (yeast extract). The yeast extract improved the production of EPS in *L. plantarum* NTMI20 too. From the inorganic nitrogen sources tested at a level of 2 g/L in the case of *L. plantarum* NTMI20, the ammonium nitrate proved to be the least efficient (about 0.09 g EPS/L), whereas ammonium sulfate was the most efficient (0.35 g EPS/L). The influence of the incubation time on EPS production was also studied, being emphasized that for both strains the highest concentration of EPS was reached after 72 h (about 0.34 g/L in *L. plantarum* NTMI05 and 0.27 g/L in *L. plantarum* NTMI20, respectively). It was supposed that the above-mentioned incubation period was the most suitable for both enzyme activity and the metabolism rate of the polysaccharide.

To determine the optimal conditions for maximum EPS production, pH values of MRS broth from 4 to 8 were tested [114]. The EPS production by *L. plantarum* NTMI05 was about 0.14 g/L at pH 4 and increased to 0.35 g/L at pH 7. A decrease in EPS concentration until 0.24 g/L was determined at pH 8. If the influence of pH on EPS production by *L. plantarum* NTMI20 is discussed, the authors reported values ranging from about 0.13 g/L at pH 4 and 0.32 g/L at pH 7. Similarly, increasing pH until 8 affected negatively EPS production by both strains. Considerably increasing EPS yield was reported as a result of pH-controlled conditions [116].

The influence of various cultivation conditions on the EPS yield by different LAB is presented in Table 1.

### 5.5. Strategies for Improving EPS Production by LAB

Two strategies were proposed to optimize and increase, both in situ and ex situ, the EPS production by LAB, in order to sustain their technological applications [52]. Thus, metabolic engineering and optimization of the production conditions, as well as the regulatory mechanisms were applied. The composition of the culture medium was identified as one of the most important factors affecting EPS production [100]. In turn, the synergic action as a result of the concomitant presence in the medium of different LAB strains is of high significance in terms of EPS production. If the complexity of synthetic mechanisms can be hindering, creating environmental stress [52] will induce the overexpression of genes related to EPS production (i.e., in species such as *S. thermophilus* and *Lcb. paracasei*) [100] and, thus, the improvement of the production of EPS with targeted functionality.

## 6. Processing Methods of Biopolymers Produced by LAB

To improve the functional properties the biopolymers produced by LAB can be modified by phosphorylation, sulfonation, and acetylation. Phosphorylated EPSs and sulfated EPSs exhibit better superoxide and hydroxyl radical scavenging ability, respectively an increased antioxidant activity. Sulphonated EPSs have a stronger inhibitory effect on Gram-positive and Gram-negative pathogens. Acetylated groups confer EPSs more flexible, elastic, antioxidant, and thermo-reversible properties [16].

Improved properties are also obtained by combining two biopolymers. Thus, PLA with poly(3-hydroxybutyrate) (PHB) films demonstrated a good barrier to water vapor [117]. EPSs composed of manan (produced by *Weissella confusa* MD1) and EPSs composed of glucose, galactose, mannose, and arabinose (produced by *Lact. fermentum* S1) have significant activity against food-borne pathogens [118].

In order to protect food and extend shelf life, food packaging/edible coating must have a number of physical, chemical, and functional properties; namely, to provide a barrier for water vapor and oxygen, to be permeable to CO_2_, to have good mechanical properties [117], to present antioxidant and antimicrobial capacity [118], and to be flexible [119], transparent, and biodegradable [117].

In order toimprov their properties, the biopolymers secreted by LAB are subjected to various processing, as follows:for higher flexibility, plasticizers that have the ability to increase the mobility of biopolymer chains due to the reduction of intermolecular forces are added. Thus, to improve flexibility the kefiran films are plasticized with sorbitol, galactitol [119], glycerol, oleic acid, polyols, and sugars (glucose, galactose, sucrose) [117], and levan films, with glycerol [120].to present effective barrier properties against water vapor and oxygen, glucose, glycerol, oleic acids [117], or fish gelatin are added to biopolymers [120];to ensure pH and high-temperature stability, EPSs are combined with biosurfactants (lipoproteins, polysaccharide-lipid complex, phospholipid), and PLA with cellulose [117];enhanced mechanical properties can be achieved when composite films made of EPSs, lipids, and hydrocolloids are formed [118]. Moreover, EPSs combined with starch (corn starch, cassava starch) form films with improved mechanical and chemical properties [117], and nanocomposite films composed of starch/kefiran/ZnO [121] or levan and starch have increased tensile strength [118].to obtain composite films with improved antioxidant properties, sodium carboxymethylcellulose [117] is incorporated into EPS-based films, or 1,3-propandiol into dextran- and chitosan-based films [122];for improved antimicrobial properties, nanocomposite films are formed by adding essential oils and other active compounds [117].

## 7. Applications of Biopolymers Produced by LAB in the Food Industry

Consumers prefer food that is healthier, with high quality and safety. In this sense, food packaging is used to effectively extend shelf life, preserve nutrients and reduce microbial contamination during food transport and storage.

The overuse of conventional plastics in food packaging contributes to multiple environmental challenges such as natural resource depletion, waste generation, and global warming [123,124,125,126]. The generation of synthetic polymer waste has increased at a worrying rate. Studies have shown that less than 10% of the synthetic plastics generated are to be recycled, raising serious concerns about the production of synthetic polymers.

Given the growing concern about the environmental impact of food packaging waste, sustainable and eco-friendly packaging is widely used to minimize the harmful effects on the environment. Biomaterials are derived from sustainable and renewable biomass, compared to finished petrochemical products [127,128,129].

The use of biopolymers is an advantageous method for replacing synthetic polymers in the concern for environmental awareness. One reason for the limited market penetration of bio-based plastics, apart from the higher price level, is represented by ecological concerns in connection with resource extraction [126]. To date, the use of biodegradable polymers such as polylactic acid (PLA), polyhydroxyalkanoates (PHAs), such as poly(3-hydroxybutyrate) (PHB), poly(3-Hydroxyvalerate) (PHV) contributed to reducing environmental damage [130].

### 7.1. Applications of PLA Produced by LAB in the Food Industry

Polylactic acid (PLA) is a compostable bioplastic [131], a material derived from natural renewable sources, made by polymerizing lactic acid monomers derived from the fermentation of starch as feedstock. Various polymer products can be manufactured using PLA, making it a tempting substitute for petroleum-based materials [132,133]. Thepermeability coefficients of CO_2_, O_2_, N_2_, and H_2_O, as well as the barrier properties against organic permeation for PLA, are comparable to petroleum-based polymers [131]. Compared to commercial polymers, PLA produces less smoke, has a lower specific weight, and melts at a lower temperature [133].

PLA is an economically useful, biodegradable natural product [134] that can be used mainly to make food packaging containers and foils (dry products and perishable products such as fruits and vegetables) [126]. It is also marketed for disposable packaging applications such as bottles, cold drink cups, containers with thermoformed trays and lids, blister packaging, overwrap packaging as well as flexible films [132,135].

PLA can be used in a variety of applications due to its ability to be thermally and stress-crystalized, copolymerized, and modified. Due to its outstanding organoleptic characteristics, PLA is considered an alternative product for food contact packaging [132]. PLA produced from biomass or agricultural waste can act as a CO_2_ sink, helping to reduce greenhouse gas emissions in the long term. The low degradation rate of PLA is affected by many variables, such as crystallinity, purity, molecular weight, temperature, and pH [133].

In order to reduce the costs associated with the fermentative production of LA for PLA synthesis, cheaper and more environmentally sustainable raw materials were used. In this context, yeast extract is used as the nitrogen source, and corn steep liquor as the carbon source. Several low-cost sources of biomass have included whey, molasses, starch materials, lignocellulose hydrolysates, and wheat bran. Different lactobacilli species (e.g., *Lb. helveticus*, *Lb. delbrueckii* subsp. *bulgaricus*, *Lb. acidophilus*, and *Lb. casei*) have been employed in LA production from whey.

LA-producing bacteria such as *Lb. lactis*, *Propionibacterium*, *Lb. delbrueckii* and *Cupriavidus necator* have also been used in a co-culture fermentation system to obtain 3-hydroxybutyrate. The use of co-cultures is beneficial due to the possibility of using by-products (e.g., whey, molasses), which are cheaper than glucose, as substrates for PHA production. These fermentations have an increased yield and improved product quality control [8].

The benefits of using PLA for food packaging applications are reflected in the fact that it is compostable under industrial conditions, produced from renewable sources, biocompatible, recyclable, and has the potential to substitute conventional plastic materials [123,133,136]. PLA has also been approved by the U.S. Food and Drug Administration (FDA) as safe for consumption. The advantages of using PLA in food packaging are reflected in its ease of processing, superior transparency, and environmental characteristics. However, it also has some disadvantages, such as sensitivity to thermal degradation, poor barrier properties to oxygen, and other gases that are influenced by the degree of crystallinity, and low mechanical performance, which hinders its industrial exploitation [126,136,137]. At present, bio-based food packaging is largely used for shorter shelf-life products that do not require high resistance to oxygen or water vapor transmission, such as fresh juices, dairy products, fruit, vegetables, and meat [138].

The above limitations reduce the wider application of PLA in food packaging and lead to studies related to the improvements of PLA properties by the addition of nanoparticles and plasticizers and by the application of advanced processing techniques or treatments [139].

Combining PLA with other compounds brings small changes in chemical composition and molecular characteristics, thus allowing wider use of PLA for various food packaging applications as well as for meeting the requirements of different food products. An environmentally friendly and cheap way to package products that are safe for humans and other living organisms is to use PLA blended with starch extracted from tapioca [140]. Starch-based biopolymers are renewable, biodegradable, and oxygen permeable, making them a good alternative for commercial packaging [133,141].

Improving PLA performance in commercial packaging applications could also consist of copolymerization with other biopolymers [142] or melt blending the PLA matrix with another high-crystallinity biopolymer matrix of similar melting temperature [135,143].

### 7.2. Applications of PHAs Produced by LAB in the Food Industry

Polyhydroxyalkanoates (PHAs) are microbial polyesters obtained by bacterial fermentation [144] with pure microbial cultures grown on various renewable sources. These are potential substitutes for conventional plastics due to their advanced biodegradability and similar physicochemical properties. Although they have comparable characteristics to ordinary plastics, the extensive use of PHA is still hampered by their high production cost [126,145,146].

Depending on the microorganism, carbon source, and growth conditions used, the distribution of monomers in the polymer and the length of the polymer chain are different. PHAs are linear thermoplastic polymers of hydroxyalkanoic acids (HA) connected by an ester linkage, which can be produced by many microorganisms as intracellular carbon and energy stores [126,147].

Among the PHAs, poly-3-hydroxybutyrate (PHV), poly-4-hydroxybutyrate (PHB), poly(3-hydroxybutyrate-co-3-hydroxyvalerate) (PHBV), and poly(3-hydroxybutyrate-co-3-hydroxyhexanoate) [14,58] are the most commercially known, whose physical and mechanical properties are very similar to those of traditional plastics [126].

In this context, polyhydroxyalkanoates (PHA) were obtained by microbial fermentation of agro-industrial by-products such as scotta and Toma cheese whey. For the metabolism of more complex low-cost substrates such as molasses, starch waste, or whey, mixed cultures of lactic-acid-producing bacteria (MMC, derived from activated sludge) such as *Lactococcus lactis* or *Lactobacillus delbrueckii* and *C. necator* were used [64].

PHAs are microbial polyesters that can be extruded into films, foils, and diaphragms with excellent moisture and oxygen reduction properties [148]. Thus, they are used in the preparation of water-resistant cardboard boxes and are considered to be an alternative to aluminum foil which is not biodegradable. 

As thermoplastic, brittle, crystalline, elastomers, flexible, hydrophobic polymers [145], they can be used to produce disposable food containers and utensils [126,145,149], as well as to obtain flexible packaging for foods with high oil content, such as marinated olives, cheese, and nuts [150].

These biopolymers can be used to produce bioplastics with high melt strength, suitable for low heat deformation during thermoforming. They are suitable for a wide range of packaging applications, including hot and cold cups, cup lids, yogurt containers, tubs, trays, and single-serve food packaging. Due to the hydrophobic nature of polyesters, PHA films present very high water-vapor barrier properties, close to those of low-density polyethylene (LDPE). PHBV is suitable for heat shaping and production of flexible plastic bags used in the food industries [151].

The development of new biopolymers of interest for wider applications in food packaging can be achieved by designing and creating new properties through the controlled synthesis of active compounds. To improve the functionality of polymer films, especially in food packaging applications, either as coatings or as shells, antimicrobial substances such as bacteriocins or silver and copper nanoparticles can be incorporated in PHAs [148].

A summary of PLA and PHAs applications in the food industry and packaging is mentioned in Table 3. It is predicted that the above-mentioned techno-economic challenges in this area will be surpassed and the food packaging market based on these biopolymers will increase, with positive effects on environmental protection and consumer acceptability.

### 7.3. Applications of EPS Produced by LAB in the Food Industry

Exopolysaccharides produced by lactic acid bacteria have gained a special interest in the food industry due to their ability to improve the rheological properties of foods, especially fermented foods, being considered natural biothickeners, as well as natural functional food ingredients [18,22,72].

Bacterial EPSs can be produced ex situ through controlled fermentation when a high-quality reproducible product added as an additive/ingredient to food, or in situ [23,24,33]. Ex situ production is an easier-to-control alternative than production in situ and with definite results in terms of the monitored characteristics of EPSs [23,40]. Instead, the use of EPSs produced in situ is a viable alternative for replacing classic additives and in order to obtain foods with a “clean label” [33,94].

There is a wide variety of EPSs differentiated by the diversity of monomer composition, chain length, degree of branching, molecular mass, three-dimensional conformation, and electrical charge. These specific characteristics confer to EPSs numerous functional properties, such as the ability to form viscous solutions, gels, films, and emulsions, to prevent syneresis, and also to sweeten [33,40,99]. On the basis of their specific properties some EPSs exhibit the ability to form intermolecular associations [26,43] and to interact with proteins [18,26,33,99]. Moreover, EPSs provide food products firmness, creaminess, and mouthfeel [33]. 

The functional characteristics of EPSs listed above make these polymers useful in the technology of obtaining food products [43].


**The ability to form viscous solutions**


EPSs have the ability to bind water, which leads to a decrease in the fluidity or an increase in the viscosity of the solutions. The viscosity of solutions is influenced by HoPSconcentration [40,153], chain stiffness [40], molecular weight [99], chain length, structure [153], the radius of gyration (Rg) [21,72], the presence of ionizable groups which gives a polyelectrolytic behavior to the biopolymer [154], and also by the complexity of the side chains [99].

Depending on the concentration, Hundschell et al. observed different behaviors of EPSs [40]. Thus, at concentrations lower than the critical concentration the biopolymers do not react with each other, while at higher concentrations, the polysaccharide chains interact with each other and interpenetrate. This interaction leads to increased viscosity.

EPSs with larger volumes [99], high molecular mass, and stiff chains lead to high viscosity, whereas small molecules and flexible chains lead to low viscosity of solutions. The exception is levan, which has a low viscosity even at a high molecular weight, perhaps due to its compact and spherical structure [40].

Juvonen et al. studied the vascularity of EPSs as an expression of texture. Thus, the branched β-glucan, produced by *Pediococcus claussenii* or *Lc. mesenteroides* E-093126, created a coarse elastic texture, compared to the weakly branched dextran, produced by *Lc. lactis* E-032298 and *W. confusa* E-90392, which created a coarse but less elastic texture [153]. These results are consistent with the fact that branched β-glucan is a powerful viscosimetric agent even at low concentrations [155];conversely, dextrans containing α-glycosidic bonds with both single unit and elongated branches form compact structures with effect on viscosity at high concentrations [28,41,153].

Nachtigall et al. and Ruas-Madiedo et al. stated that molar mass and radius of gyration are correlated with the thickening effect of EPSs in dairy products [72,95].

Negatively charged EPSs increase the viscosity of the products because they can interact with proteins at the isoelectric point, through electrostatic forces, forming a continuous branched network [99]. In general, polysaccharides with high viscoelastic properties are produced by strains of *L. lactis* subsp. *cremoris*, *Lb. rhamnosus*, *Lb. casei*, *Lb. helveticus* and *S. thermophilus* [99].

Due to their ability to produce viscous solutions, EPSs are used as thickening agents.


**The ability to form gels**


Concentrated solutions of exopolysaccharides (for example, over 3% for curdlan), maintained for a long time at high temperatures, due to high molecular interactions, can form a triple helix cross-linked network stabilized by hydrophobic interactions, which upon cooling forms a stable gel that is thermoirreversible [156] A stable gel is obtained if several cross-links are formed inside it, as in the case of EPSs with a high molecular mass [157]. Moreover, soluble EPSs containing β-glycosidic bonds with regular structures can form gels [40]. The firmness of gels is positively correlated with the number of side chains, the presence of branches, and Rg values [99].

In general, polysaccharides that contribute to gel stiffness are produced by *L. lactis ssp. cremoris* [99].


**The ability to prevent syneresis**


Syneresis is a phenomenon that can occur in fermented milk products as a result of the destabilization of casein micelles at low pH values [33]. Reduction of syneresis is a common property of different EPSs [99]. Producing EPSs in situ solves this problem better than adding them as ingredient [33]. A positive correlation was observed between syneresis reduction and the presence of rigid chains, or neutral EPSs [99].


**The ability to influence taste**


Certain strains of lactic acid bacteria (for example *Lc. citreum* and *Lc. mesenteroides*) characterized as heterofermentative, are able to produce mannitol directly from fructose. These strains use fructose as an electron acceptor and reducing it to mannitol, reaction catalyzed by mannitol 2-dehydrogenase (MDH), without producing sorbitol as by-product. Acetic acid and mannitol are substances that influence the taste and smell of the food product. Juvonen et al. fermented carrot puree with the two strains of lactic acid bacteria and obtained a product with a strong acid taste and odor [153].


**The capacity to form films**


EPSs have the ability to form films, which is why they are used in the food industry to obtain biodegradable and edible food packaging, emulsions, and encapsulated products [7,158]. Exopolysaccharides from LAB (dextran, levan, kefiran, and hyaluronic acid) can form films that improve the quality and extend the shelf life of food commodities [7].

These biopolymers cannot be used as such but only together with other substances (plasticizers) because of their ability to absorb water (they are hydrophilic), the fragile films formed, and their poor mechanical properties. For example, kefiran forms a rigid, brittle, transparent, and hydrophilic film, but in the presence of substances with a plasticizing role (i.e., glycerol, sorbitol [7], polyols, nanomaterials) its water vapor permeability is reduced [158].


**Application of exopolysaccharides in the food manufacturing**


EPSs produced by lactic acid bacteria have numerous applications in the food industry, especially in fermented products, where they are formed in situ. Moreover, they can be added as ingredients/additives when produced ex situ.

EPSs are involved in the production of food such as: fermented dairy products (i.e., yogurt, cheese [16], low-fat cheese [159], low-calorie curd, kefir [160]); bakery products (i.e., bread [161], sourdough, gluten-free products [162]) and fermented vegetable products (i.e., pureed [153], yogurt from cereals [163], fermented beverages from cereals [164], protein concentrate [37], and vegetable doughs [97].

Recent food applications of exopolysaccharides produced by lactic acid bacteria and their technological properties are summarized in Table 4.


**Application of exopolysaccharides in the packaging industry**


Food packaging protects the foods from biological, chemical, and physical contaminations [117]. The use of EPSs to obtain food packaging is a topic of interest [7,158]. Food can be protected using ”food films” or ”food coatings” [94] with resistance to oxygen moisture [158], ultraviolet rays [121], and microorganisms [118]. The “food film” is a thin film applied on the food product to protect it. “Food coatings” are suspensions or emulsions that can be applied directly to the surface of food by spraying or dipping. They solidify and form an edible film with low water vapor permeability and good mechanical properties [120] which can preserve the taste, aroma, texture, and appearance of food [120,158].

To obtain packaging with improved properties, EPSs are combined with plasticizers (glycerol, sorbitol [7], polyols [158]), or other components (e.g., starch/ZnO [121], carboxymethyl cellulose/copper oxide nanoparticles [172]) to form composite/biocomposite/nanocomposite packaging [120]. These films have a smooth, glossy surface and good structural integrity [94,117].

In this sense, kefiran is an EPS with potential applications in food packaging. To obtain packages with improved properties, kefiran is combined with plasticizers or enters the structure of composite packages. Thus, Zolfi et al. have synthesized a packaging film from kefiran, whey protein isolate, and montmorillonite which presented an increased tensile strength and a low water vapor permeability [173].

Davidović et al. al have plasticized dextran, produced by *Lc. mesenteroides* T3, with sorbitol in order to obtain edible coatings useful in protecting food (fruits and vegetables) [174].

In the presence of some ingredients, kefiran, dextran, and levan have the ability to form nanocomposite films. These films are used in the food industry to obtain food packaging or edible packaging [94]. Babaei-Ghazvin et al. have obtained a biodegradable nanocomposite packaging film composed of starch/kefiran/ZnO with UV protection, which prevents the degradation of sensitive-to-light nutrients, lipids photo-oxidation, food discoloration, and flavorquality loss [121].

Kefiran-based films, due to the presence of nisin, a natural antibiotic with antibacterial and antimicrobial properties, offer protection to food from microorganisms, improving their safety [158]. EPSs produced from *Lactococcus lactis* F-mou strain, due to their antibacterial activity, can be used in the form of coating or films to obtain antimicrobial packaging [118].

Li et al. developed an edible film from cassava starch with sodium carboxymethylcellulose and glycerol in which they incorporated two large EPS-producing LAB strains (*Lactobacillus plantarum* and *Pediocococcus pentosaceus*) [6]. This film extended the shelf life of bananas because it created a barrier to water and light and showed good antioxidant activity.

Therefore, EPSs produced by LAB can be used for the production of new food coatings to replace environmentally unfriendly packaging [94].

Due to the fact that they are produced in small quantities by LAB, EPSs are not widely studied and have few applications in food packaging. The EPS producing lactic acid bacteria, the type of packaging obtained by using EPS, and their functional properties respectively are shown in Table 5.

## 8. Concluding Remarks and Outlook

Lactic acid bacteria have a long history of safe application in fermentation processes and are exploited in industrial processes as starters for obtaining ripened cheeses and meat products, fermented dairy products or pickled vegetables, and recently as probiotics. The demand for biopolymers is increasing both due to increasing costs of the raw materials for plastic production and to the tendency at different levels to achieve the principles of the circular bio-economy.

A wide range of renewable resources (feedstocks such as wheat, sugarcane, lignocellulosic waste, food waste, etc.) can be used for biopolymers’ production. Food waste, including kitchen residues, has more potential for lactic acid production due to its high carbohydrate content and, furthermore, does not require expensive pre-treatment.

LAB proved to have characteristics and to produce metabolites that recommend numerous bacterial strains not only for expanding the range of traditional fermentation but also for producing biopolymers with high yields and productivity. Thus, lactic acid as the main product of lactic acid fermentation is considered a versatile green platform compound, widely used in the production of polylactic acid. An increasing application of PLA as bioplastic leads to the recent increase in its demand. Quite recently, the synthesis of polylactic acid directly by fermentation, based on genetic manipulation of microorganisms, opens the journey of PLA applications in fields not very well deepened, due to the improved features of this biopolymer and the associated reduced costs of production

PHAs are green materials, promising candidates to replace petrochemical plastics on the basis of their biodegradability. Used extensively in other fields than the food industry (e.g., in the medical field), they have the potential to be used in the production of films, packaging foils, containers, and single-serve food packaging. Although the production costs are high compared to conventional petroleum-based plastics, PHA growth and expansion could be sustained by developing co-culture fermentation systems in which LAB strains occupy an important position.

Novel applications of EPS-producing LAB strains, traditionally met in the manufacture of dairy products and other food as providers of thickening agents, were underlined, including those in food packaging. Screening of EPS, development of methods of EPS quantification and characterization in terms of both monomer composition and linkages between them, and last but not least designing strategies of improving EPS production by LAB should be the focus of future research.

The biodegradability of biopolymers needs to be considered along with their popularity. Dispersal in various environments will be the consequence of the biopolymers’ overproduction and overuse. In the case of PLA, which is partially degradable, the possibility of its environmental accumulation is increasing day by day. The biodegradability of bioplastics can be assessed in present through several test methods and only in some environments.

Finally, the biosynthesis and the use of PLA, PHAs, and EPSs in the food industry including food packaging are strongly related to their production costs. Reducing the costs and improving the yield by using food waste in the production of biopolymers is challenging, but a deep knowledge of the processes as a whole (i.e., management of the fermentation conditions, optimization of the utilization rate of substrates, the downstream steps improvement, improving the industrial adaptability of LAB strains by biological engineering, etc.) will sustain the overcoming of these challenges and will contribute also to the true implementation of the circular economy concepts and environment protection towards the European Green Deal.

## Figures and Tables

**Figure 1 polymers-15-01539-f001:**
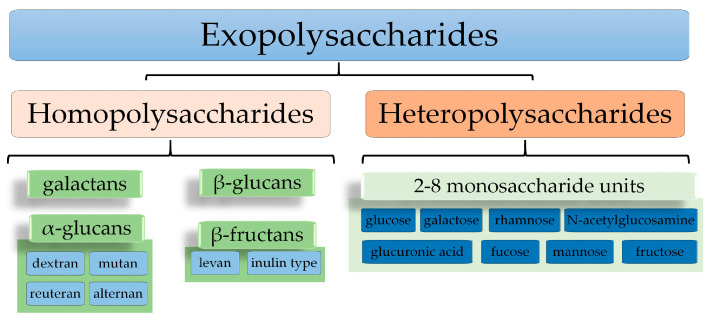
Classification of exopolysaccharides produced by lactic acid bacteria.

**Figure 2 polymers-15-01539-f002:**
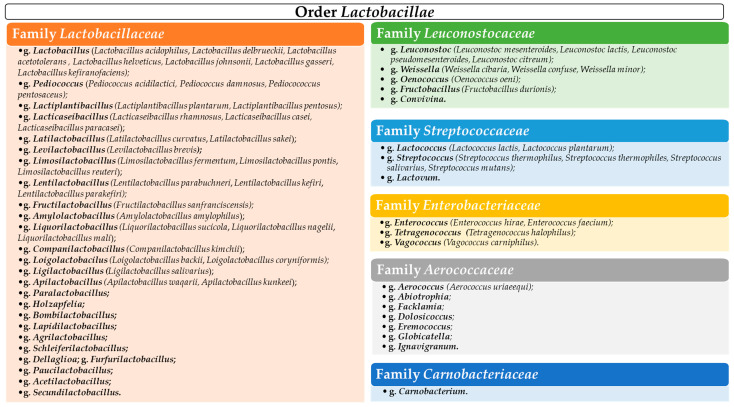
Lactic acid bacteria taxonomy.

**Figure 3 polymers-15-01539-f003:**
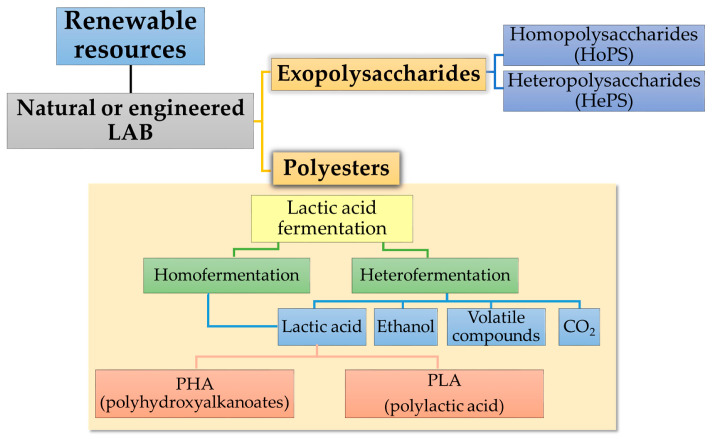
Polymers produced by lactic acid bacteria—general overview.

**Figure 4 polymers-15-01539-f004:**
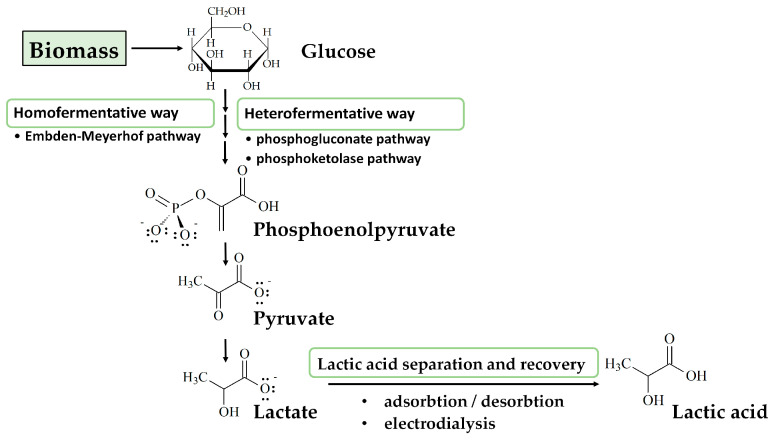
Lactic acid production by LAB fermentation.

**Table 1 polymers-15-01539-t001:** Selected studies on biopolymers produced by lactic acid bacteria in a different environment.

Biopolymers Producing LAB	Carbon Substrate	Biopolymers Produced	Experimental Conditions	Scale	Measurement	Parameters of Biopolymers Production	Ref.
Mixed microbial cultures (MMC) derived from dairy-activated sludges	Scotta cheese whey (non-treated)	PHA	Temperature 30 °C, incubation in the dark, pH not regulated, rotary shaking at 120 rpm, duration 48 h	Flask	PHA extraction from dried biomass according to method of chloroform-hypochlorite dispersion	PHA yield 0.35 g/g; productivity 0.0088 g/L/h	[64]
MMC	Scotta cheese whey (non-treated)	PHA	Temperature 30 ± 0.2 °C, pH 7 ± 0.02, agitation 120 rpm, duration 29 h	10 L Bioreactor		PHA yield 0.52 g/g; productivity 0.037 g/L/h	
MMC	Toma cheese whey (pre-treated by thermo-calcic precipitation and ultrafiltration)	PHA	Temperature 30 ± 0.2 °C, pH 7 ± 0.02, agitation 120 rpm, duration 24 h	10 L Bioreactor		PHA yield 0.37 g/g; productivity 0.018 g/L/h	
*L.mesenteroides* isolated from MMC	Modified Khardhenavis synthetic medium (acetic acid replaced with glucose)	PHA	Inoculum 10% *v*/*v*, temperature 30 °C, duration 24 h	Flask		PHA yield 0.036 g/g; productivity 0.0035 g/L/h	
*L.mesenteroides* isolated from MMC	Modified Khardhenavis synthetic medium (acetic acid replaced with lactose)	PHA	Inoculum 10% *v*/*v*, temperature 30 °C, duration 48 h	Flask		PHA yield 0.036 g/g; productivity 0.00119 g/L/h	
*Lactobacillus plantarum* CW10 isolated from fermented cow milk	PHA screening on Medium A2 containing starch, yeast extract, peptone PHA production in medium E, containing (NH_4_)_2_HPO_4_, K_2_HPO_4_, MgSO_4_ and carbon source	PHA	Variable/optimized	Flask	PHA extraction with chloroform; PHA characterization using FTIR; microstructure and surface morphology characterization of PHA using SEM	PHA yield: 25.3% in liquid medium A2; 14.7% in medium E supplemented with ammonium sulfate; 21.6% at pH 6.0; 24.4% at temperature by 35 °C; 20.5% at agitation speed by 200 rpm	[14]
*Lactobacillus* *casei* WWD3 isolated from dairy wastewater		PHA	Variable/optimized	Flask		PHA yield: 16.1% in liquid medium A2; 14.1% in medium E supplemented with ammonium sulfate; 16.4% at pH 6.0; 15.7% at temperature by 40 °C; 19.7% at agitation speed by 200 rpm	
*Lactobacillus* species isolated from food produced in Turkey	MRS (de Man, Rogosa, and Sharpe) broth	PHB	Inoculum 2% (*v*/*v*); incubation 48 h at optimal growth temperature of each strain (30 ± 1 °C, 37 ± 1 °C and 40 ± 1 °C, respectively)	Flask	PHB extraction with hot chloroform; PHB crystals converted into crotonic acid and absorbance measured spectrophotometrically (235 nm)	PHB yield: 35.8% (*L. bulgaricus* C8); 29.4% (*L. brevis* C3); 29.0% (*L. casei* C5); 17.1% (*L. acidophilus* C4); 13.8% (*L. plantarum* (A-C1))	[68]
*Lactococcus* species isolated from food produced in Turkey	MRS or Elliker broth medium	PHB	Inoculum 2% (*v*/*v*); incubation 48 h at 30 ± 1 °C	Flask		PHB yield: 20.9% (*L. lactis* A1); 18.5% (*L. cremoris* A3)	
*Pediococcus* species isolated from food produced in Turkey		PHB	Inoculum 2% (*v*/*v*); incubation 48 h at 37 ± 1 °C	Flask		PHB yield: 8.0% (*P. halophilus* B6)	
*Streptococcus* species isolated from food produced in Turkey		PHB	Inoculum 2% (*v*/*v*); incubation 48 h at 40 ± 1 °C	Flask		PHB yield: 17.2% (*S. thermophilus* E1)	
*Streptococcus thermophilus* strains from culture collection and commercial providers	Semi-defined medium with lactose (L-SDM) in batch fermentation of ST-143; UHT milk fortified with low-heat skim milk powder for screening of gelation and EPS production of *S. thermophilus* strains	Free and capsular EPS	Isolation and purification of EPS: temperature 43 °C; constant pH 6.0 during fermentation; agitation at 200 rpm; anaerobic conditions Acidification of milk: 10 g L^−c^ starter, temperature 43 °C; fermentation until pH 4.6	Bioreactor	Isolation of free EPS by using trichloracetic acid; heating step applied for detaching capsular EPS from cells; EPS quantification by phenol-sulphuric acid method; ion exchange chromatography for determination of the EPS charge	EPS concentration ranged from 2 ± 2 mg kg^−m^ (strain DSM20259) to 138 ± 24 mg kg^−^ (strain DGCC2057); ratio free: capsular EPS approx. 4:1	[69]
*Streptococcus thermophilus* CC30 isolated from raw milk	Skim milk	EPS	Incubation 24 h at 30 °C	Flask	EPS precipitation with cold absolute ethanol; monosaccharide composition determined by GC; thermogravimetric analysis of EPS; SEM applied for EPS microstructure and surface morphology; AFM images of EPS; reducing power of EPS	EPS yield 1.95 g/L; EPS molecular weight 58–180 KDa, with glucose and galactose as monomers, in molar ratio 1.4:1.6; EPS of pseudoplastic nature, with spherical structure; conformational spike-like lumps with height ranged from 10 to 30 nm revealed by AFM; moderate antioxidant capacity	[70]
*Streptococcus thermophilus* 05-34 isolated from Tibetan kefir grains	Reconstituted skim milk (RSM)	EPS	pH 5.0, 6.0, 6.5 and 7.0, respectively; growth temperature 27 °C, 30 °C, 37 °C and 40 °C, respectively; fermentation time 8 h, 16 h, 24 h, 32 h, and 40 h, respectively; glucose, sucrose, galactose, lactose, and fructose as carbon sources; tryptone, soy peptone, peptone, beef extract, and whey protein concentrate as nitrogen sources	Not mentioned	EPS concentration determined by the phenol–sulfuric acid method; HPLC system for determination of the molecular mass of EPS; Real-time quantitative PCR for determination of the effect of different fermentation conditions on the expression level of epsCin *S. thermophilus* 05-34	EPS concentration ranged depending on the conditions of fermentation: 250 mg/L (soy peptone of 30 g/L); 120 mg/L (sucrose concentration 80 g/L); 100.25 mg/L (37 °C, pH 7.0, time 30 h); optimal fermentation condition: 10% RSM with 80 g/L sucrose and 30 g/L soy peptone at initial pH 7.0 and 37 °C for 30 h; molecular mass of EPS (obtained under the optimal conditions) 4.7 × 10^5^ Da; monosaccharides composition of EPS: galactose and glucose (molar ratio 1.0:0.8), mannose, rhamnose	[71]
*Streptococcus thermophilus* DGCC7919 (commercial culture)	Whey permeate medium	EPS	Agitation rate of 200 rpm; anaerobic conditions; temperature 40 °C; pH 6.0	5 L bioreactor	EPS quantified photometrically using the phenol sulphuric acid method; chemical composition of EPS by HPAEC-PAD after acid hydrolysis; molecular mass of EPS by size exclusion chromatography coupled with an RI detector; structure elucidation by GC–MS	Total EPS 404 ± 8 mg GE/L (GE–glucose equivalent); composition of EPS: glucose, galactose, rhamnose, glucosamine	[72]
*Streptococcus thermophiles* S-3 CGMCC 12098	Skim milk	EPS	Fermentation at 42 °C for 24 h under anaerobic conditions	5 L bioreactor	Molecular weight distribution by HPSEC; NMR analysis for partial structural analysis; monosaccharide compositions by HPAEC	EPS yield up to 100 mg/L; composition of EPS: N-acetyl-galactosamine, galactose and glucose in the molar ratio of 1:2:1; molecular weight 574.0 kDa	[73]
*Weissella cibaria* and *Weisella confusae* isolated from African fermented milk and cassava products	MRS media supplemented with sucrose or raffinose and glucose	EPS	Incubation 96 h at 30 °C; anaerobic conditions	Microplates; agar plates	Acid hydrolysis for determination of monomer composition; NMR spectroscopy for the chemical shift assignment of EPS; HP-GPC for EPS molar mass estimation	EPS type (related molar mass): dextran (>2 × 10^7^ g/mol); levan (2 × 10^5^ g/mol); inulin (1.9 × 10^7^ g/mol); HePS (not determined), depending on selected *Weissella* isolates	[67]
*Weissella cibaria* strains C43-11 and C2-32, isolated from wheat semolina	Modified MRS broth (MRS with yeast extract and maltose added), supplemented or not with sucrose	EPS	24 h culture inoculated at 4% (*v*/*v*); incubation 24 h at 30 °C	Flask	EPS concentration determined according to the phenol-sulfuric method; EPS monosaccharide composition determined by HPAEC-PAD	EPS production by 0.13 ± 0.00 g/L (strain C2-32) and 11.74 ± 3.25 g/L (strain C43-11), respectively, in MRS with sucrose; EPS composed by glucose (125.8 ± 22.7–16, 299.9 ± 2114.1 mg/L), mannose (1245.9 ± 10.6–280.5 ± 23.3 mg/L), and fructose(13.8 ± 4.7–779.9 ± 2.7 mg/L), respectively	[74]
*Weissella cibaria* C43-11 isolated from wheat semolina	Modified MRS broth (MRS with yeast extract and maltose added) Liquid sourdough formulation (wheat flour with wheat gluten/amaranth flour/quinoa flour; dough yields (DY) of 250 and 500, respectively; with or without sucrose)	EPS	Screening for EPS production: inoculation at 4% (*v*/*v*) in mMRS with added sucrose; incubation for 24 h at 30 °C Production of EPS in liquid sourdoughs: inoculation at 4% (*v*/*v*) in the flour mixtures and incubation for 15 h	Plates Flasks	EPS concentration determined according to the phenol-sulfuric method	EPS production in synthetic media: 18.56 ± 2.64 g/L; higher EPS production (20.79 ± 3.55 g/kg LS) (formulation with quinoa flour and 6% sucrose at DY 250)	[75]
*Weissella minor* W4451 isolated from sourdough	Semidefined medium (SDM) for bacteria cultivation Skim milk powder reconstituted at 9% (*w*/*v*)	EPS	2% overnight subcultures used for milk fermentation at 37 °C until pH 4.5 (8 h)	Bioreactor	Total EPS isolation from SDM by double cold ethanol precipitation; EPS characterization by GC-MS; EPS isolation from fermented milk with trichloroacetic acid; polymers’ quantification as polymer dry mass	Monosaccharides composition of EPS: glucose (57.0 ± 5.1%), rhamnose (12.4 ± 1.1%), mannose (12.6 ± 1.1%), ribose (0.5 ± 0.5%), and galactose (17.5 ± 1.6%); EPS yield 1.58 g/L	[66]
*Lactobacillus plantarum* ITM21B isolated from sourdough	Modified MRS broth (MRS with yeast extract and maltose added) Liquid sourdough formulation (wheat flour with wheat gluten/amaranth flour/quinoa flour; dough yields (DY) of 250 and 500, respectively; with or without sucrose)	EPS	Screening for EPS production: inoculation at 4% (*v*/*v*) in mMRS with added sucrose; incubation for 24 h at 37 °C Production of EPS in liquid sourdoughs: inoculation at 4% (*v*/*v*) in the flour mixtures and incubation for 15 h	Plates Flasks	EPS concentration determined according to the phenol-sulfuric method	EPS production in synthetic media: not detected; higher EPS production (4.61 ± 0.01 g/kg LS) (approx. 11 g/kg of flour, in formulation with wheat flour: quinoa flour = 1:1 and 3% sucrose at DY 250	[75]
*Lactobacillus delbrueckii* subsp. *bulgaricus* 2214 from culture collection	Semidefined medium (SDM) for bacteria cultivation Skim milk powder reconstituted at 9% (*w*/*v*)	EPS	2% overnight subcultures used for milk fermentation at 37 °C until pH 4.5 (9 h)	Bioreactor	Total EPS isolation from SDM by double cold ethanol precipitation; EPS characterization by GC-MS; EPS isolation from fermented milk with trichloroacetic acid; polymers’ quantification as polymer dry mass	Monosaccharides composition of EPS: glucose (90.0 ± 8.0%), rhamnose (9.1 ± 0.8%), mannose (0.9 ± 0.1%); EPS yield 1.88 g/L	[65]
*Lactobacillus delbrueckii* subsp. *bulgaricus* 147 from culture collection	Semidefined medium (SDM) for bacteria cultivation Skim milk powder reconstituted at 9% (*w*/*v*)	EPS	2% overnight subcultures used for milk fermentation at 37 °C until pH 4.5 (13 h)	Bioreactor		Monosaccharides composition of EPS: glucose (42.2 ± 3.8%), rhamnose (2.8 ± 0.3%), mannose (6.0 ± 0.5%), ribose (0.8 ± 0.1%), and galactose (48.2 ± 4.3%); EPS yield 0.96 g/L	[65]
*Lactobacillus delbrueckii* subsp. bulgaricus OLL1073R-1	Skimmed milk	EPS	*L. delbrueckii* incubated in 10% skimmed milk for 18 h at 37 °C	Not mentioned	Physicochemical characterization of EPS by SEC-MALS; sugar analysis by GLC-FID or GLC-MS; sugar linkage positions by GLC and MS; NMR spectrum	1546 mg of crude EPS (1260 mg of neutral EPS/10 kg of total cell culture); composition of EPS: D-Glucose, 1; D-Galactose, 1.5; molecular mass 5.0× 10^6^ g/mol	[76]
*Lactococcus lactis* LL-2A (commercial culture)	Whey permeate medium	EPS	Agitation rate of 200 rpm; anaerobic conditions; temperature 40 °C; pH 6.0	5 L bioreactor	EPS quantified photometrically using the phenol sulphuric acid method; structure elucidation by GC–MS	Total EPS (354 ± 19 mg GE/L) (GE–glucose equivalent); composition of EPS: glucose, galactose	[72]
*Leuconostoc* sp. isolated from fermented vegetables	Modified MRS (with sucrose instead of glucose)	EPS	2% fresh culture added to MRS; tested temperatures by 20 °C, 28 °C, 37 °C, 42 °C; salt concentrations tested: 1%, 3% and 5%	Agar plates	EPS presence and molecular mass estimation by gel permeation chromatography; EPS yields determined gravimetrically; monomer composition by automated thin-layer chromatography	EPS yield 25.83 g/L for *Leuconostocmesenteroides/pseudomesenteroides 406* (28 °C, 5% NaCl); 0–14.52 g/L for *Leuc. citreum* 52 and 0–8.43 g/L for *Leuc.* sp. 208, respectively (depending on the grown conditions)	[77]
*Leuconostoc mesenteroides* SN-8 isolated from Dajiang (natural fermentation)	MRS broth	EPS	Incubation at 30 °C for 48 h; shaking at 80 rpm	Flask	Monosaccharide composition by GC; molecular weight of EPS by HPSEC analysis; FTIR spectroscopy for analysis of the functional groups of purified EPS; NMR analysis of EPS; thermodynamic stability analysis by DSC and TGA; in vitro antioxidant activity; in vitro antitumor activities	Molecular weights 1.46 × 10^5^ Da; EPS contain a highly branched main chain of dextran with (1-6) linkages, and few mannose residues; DPPH radical scavenging capacity up to 57.42 ± 1.38%	[78]
*Leuconostoc mesenteroides* DRP105 isolated from Chinese sauerkraut	MRS–S medium (glucose replaced by sucrose)	EPS	Optimized conditions: sucrose 86.83 g/L, tryptone 15.47 g/L, initial pH 7.18, 36h of fermentation	Flask	Elemental analysis of EPS; total sugar content of EPS by phenol sulfuric method; chain conformation of EPS characterized by Congo red test, β-elimination, and circular dichroism	Maximum EPS yield 53.79 ± 0.78 g/L; composition of EPS: uronic acid, sugar, sulfate	[79]

*Legend:* FT-IR—Fourier transform infrared; SEM—scanning electron microscopy; NMR–nuclear magnetic resonance spectroscopy; HP-GPC—high-performance gel permeation chromatography; GC—gas chromatography; MS—mass spectrometry; GC-MS —gas chromatography–mass spectrometry; GLC—gas–liquid chromatography; AFM—atomic force microscopy; HPAEC-PAD—high-performance anion-exchange chromatography/pulsed amperometric detection; PCR—polymerase chain reaction; HPSEC—high-performance size exclusion chromatography; SEC-MALS—size-exclusion chromatography coupled with multi-angle light scattering; FID—flame-ionization detection; DSC—differential scanning calorimetry; TGA—thermal analysis.

**Table 2 polymers-15-01539-t002:** Homopolysaccarides and corresponding LAB producers.

HoPS	Sub-Divisions	Representatives	Structural Features	HoPS-Producing LAB	References
Glucans	α-glucans	dextran	α-D-Glc(1,4)	*Lactobacillus reuteri*	[18]
		dextran	α-D-Glc(1,6)	*Leuconostoc mesenteroides* *Lacticaseibacillus casei*, *Latilactobacillus sakei*, *Limosilactobacillus fermentum*, *Lentilactobacillus parabuchneri*	[5,100]
		mutan	α-D-Glc(1,3)	*Lactobacilus reuteri*	[18]
		reuteran	α-D-Glc(1,4)/α-D-Glc (1,6) with α-D-Glc (1,4)/α-D-Glc(1,6) branching points	*Lactobacilus reuteri*	[18]
		reuteran	α-D-Glc(1,4)	*Lactobacillus reuteri*	[100]
		alternan	α-D-Glc(1,6)/α-D-Glc (1,3)		
		alternan	Alternating α-D-Glc(1,6) and α-D-Glc(1,3)	*L. mesenteroides*, *Leuconostoccitreum*	[100]
	β-glucans	-	β-D-Glc(1,3) with side chain linked(1,2)	*-*	[18]
Fructans	levan-type	-	β-D-Fru(2,6)	*Lactobacilus reuteri*	[18]
		levan	β-D-Fru(2,6)	*Streptococcus salivarius*, *Streptococcus mutans* *Lactobacillus reuteri*, *Lactobacillus sanfranciscensis*	[5,100]
	inulin-type	-	β-D-Fru(2,1), being both β-fructans	*Lactobacilus reuteri*	[18]
	inulin-type	-	β-D-Fru(2,1)	*Streptococcus mutans*, *Lactobacillus reuteri*	[100]
Polygalactans			pentameric repeating unit of Gal	*Lactococcus lactis* subsp. *lactis* H414 *Lactobacillus delbrueckii* subsp. *bulgaricus* *Lactobacillus plantarum*	[18]

Glc—glucose; Fru—fructose; Gal—galactose.

**Table 3 polymers-15-01539-t003:** Applications of PHAs and PLA biopolymers produced by LAB in the food packaging industries.

Source/Substrat	Microorganisms	Biopolymers Produced	Properties of Biopolymers	Applications in the Food Packaging	References
Scotta cheese whey	Mixed microbial cultures (MMC): *Lactococcus lactis*, *Lactobacillus delbrueckii*, *C. necator*	PHA	-^1^	-^1^	[64]
Toma cheese whey and supplementedKhardhenavis synthetic media with glucose or lactose	*L. mesenteroides* isolated from MMC	PHA	-^1^	-^1^	[64]
Toma cheese whey and supplemented Khardhenavis synthetic media with glucose or lactose	Mixed microbial cultures (MMC): *Lactococcus lactis*, *Lactobacillus delbrueckii*, *C. necator*	PHA	-^1^	-^1^	[64]
Glucose and ammonium sulfate was used as carbon and nitrogen sources	*Lactobacillus plantarum* CW10 and *Lactobacillus* *casei* WWD3	PHB	-^1^	-^1^	[14]
De Man, Rogosa and Sharpe agar (MRS)	*Lactococcus*, *Lactobacillus*, *Pediococcus* and *Streptococcus*	PHB	Versatile biopolymers, with properties like conventional plastics	Production of films	[8]
Xylose, glucose	*A co-culture fermentation system: Lb. lactis*, *Propionibacterium*, *Lb. delbrueckii* and *Cupriavidus necator*	PHB			[8]
Kenaf biomass, carob pods, wheat straw, sunflower meal	Bacteria and PHB-rich biomass	PHB	-^1^	-^1^	[138]
Crude glycerol	*Enterococcus* sp. NAP11	PHB	High plasticity and accessibility to melt extrusion, injection molding, thermoforming	Production of disposable food containers and utensils; production of hot and cold cups, cup lids, yogurt containers, tubs, trays, and single-serve food packaging	[126,149,152]
Harvesting residues from food waste, sugarcane crops, bagasse, molasses, and corn stover	*Lactobacillus pentosus* and *Bacillus subtilis*	PLA	Excellent twist retention characteristics; flexible	Food packaging films; bottles, cups, containers, jars, bowls, bags	[138]
Molasses, corn syrup, whey, dextrose and cane or beet sugar	*Lactobacillus delbrueckii*, *L. amylophilus*, *L. bulgaricus*, *L. leichmanii*, *L. rhamnosus*	PLA	Thermoplastic; Renewable packaging material;	Packing dairy products, PLA-based pots for yogurt; transparent films	[148]
Corn steep liquor, whey, molasses, starchy materials, lignocellulose hydrolysates, and wheat bran (biomass)/Corn steep liquor (carbon source)/Yeast extract (nitrogen source)	*Lb. helveticus*, *Lb. delbrueckii* subsp. *bulgaricus*, *Lb. acidophilus*, *Lb. casei*	PLA	Biodegradability and biocompatibility, thermoplastic and high tensile strength, versatile and attractive for various commodities	Food and goods packaging and cutlery	[8]
Complex medium composed of free sugars (brownjuice) and starch	*Lb. plantarum A6*	PLA			[8]
Lignocellulosic hydrolysates	*Lb. brevis*	PLA			[8]

PHA: polyhydroxyalkanoate; PHB: polyhydroxybutyrate; PHBV: poly(3-hydroxybutyrate-co-3-hydroxyvalerate); PHBH: polyhydroxybutyrate–hexanoate, P4HB: poly-4-hydroxybutyrate; PLA: polylactic acid. -^1^—not specified.

**Table 4 polymers-15-01539-t004:** Applications of EPS produced by lactic acid bacteria in foods.

Food Products	Lactic Acid Bacteria	Exopolysaccharides	Food Product	Technological Properties/ Characteristics	References
Dairy products	*Leuconostoc mesenteroides*	Kefiran	Chemically acidified skim-milk gels	Improving rheological properties	[165]
	*L. fermentum* Lf2	EPS extract	Yogurt	Creamy consistency and increased hardness Improving water-holding capacity	[166]
	*Streptococcus thermophilus* zlw TM11, *Lactobacillus delbrueckii* subsp. *bulgaricus* 3 4.5	Exopolysaccharides	Yogurt	Decreasing in syneresis and improvement of texture	[167]
	*Lactobacillus plantarum* KX881772 and KX881779 EPS-producing (ropy)	Exopolysaccharides	Low-fat akawi cheese	Improving of rheological and sensory properties	[159]
	*L. lactis* subsp. PM 23, *L. lactis* NCDC191	Exopolysaccharides	Fat-free Dah	Improving texture and flavor	[160]
Bakery products	*Leuconostoc mesenteroides* ATCC 8239T	Dextran	Sourdoughrich in dextran	Increasing the volume of the bread	[163,168]
	*Weissella cibaria* MG1	Dextran	Gluten-free sourdough (from buckwheat, oat, quinoa, and teff)	Decreasing dough strength and elasticity of sorghum sourdough	[162]
	*Leuconostoc lactis* 95A and *Lactobacillus curvatus* 69B2	Glucan, dextran	Bread (30% of sourdough)	Higher volume, higher moisture content, and better mechanical properties during storage	[161]
Vegetable products	*Leuconostoc lactis* and *Weissella confusa*	Dextran	Pureed carrots	Thick texture and pleasant taste and odor	[153]
	*Weisella cibaria MG1*	Dextran	Quinoa based yogurt	Viscosity enhancement and increase in water-holding capacity	[163]
	*Leuconostoc pseudomesenteroides* DSM 20193 and *Weissella confusa* VTT E-143403	Dextran	Fava bean protein concentrate	Increasing viscosity, preventing protein aggregation stability, and improving texture	[37]
	*Leuconostoc pseudomesenteroides*, *Leuconostoc mesenteroides*, *Leuconostoc citreum*, *Weissella cibaria*, *Lactobacillus plantarum*	Dextran	Fava bean doughs	Texture modification and gel structure strengthening	[97]
	*Pediococcus damnosus 2.6* and *Lactobacillus brevis G-77*	β-glucan	Fermented beverage based on oats	Increasing viscosity and elasticity	[164]
	*Lactobacillus plantarum 90*	β-glucan	Fermented oat-based foods	Increasing viscosity	[169]
Confectionery	*Leuconostoc mesenteroides*	Alternan		Sweetener	[170]
Others	*Lactobacillus plantarum* BR2	Heteropolysaccharidecomposed of glucose andmannose	Functional foods	Improving rheological properties	[171]

**Table 5 polymers-15-01539-t005:** Applications of EPS produced by lactic acid bacteria in the food packaging industry.

Lactic Acid Bacteria	EPS	Type of Packaging	Composition of the Film	Functional Properties	Food Applications	References
*Leuconostoc mesenteroides* T3	Dextran	Edible coatings	Dextran plasticized with sorbitol	Good mechanical properties and low water vapor permeability	Food packaging	[174]
*Lactobacillus plantarum* and *Pedocococcuspentosaceus*	-^1^	Composite films/edible films	Cassava starch/sodium carboxymethylcellulose with embedded LAB	Enhanced antioxidant activity; Protection against ultraviolet light	Extended shelf life of bananas	[6]
*Lactobacillus kefiranofaciens*	Kefiran	Biocomposite films	Kefiran–carboxymethylcellulose with Saturejakhuzestanica essential oil incorporated	Increasing antioxidant properties; Inhibitory effects against *Staphylococcus aureus* and *Escherichia coli*	Food packaging	[175]
LAB from kefir grains	Kefiran	Nanobiocomposite films	Kefiran–carboxymethyl cellulose with copper oxide nanoparticles and Saturejan khuzestanica essential oil	Improvement of physical and mechanical properties; Antimicrobial characteristics against *S. aureus* and *E. coli*	Food packaging	[172]
LAB from kefir grains	Kefiran	Biodegradable edible film	Kefiran plasticized with glycerol and sorbitol	Extendible films which presented physical, mechanical, and water vapor barrier properties	Edible food films and coatings	[7]
LAB from kefir grains	Kefiran	Biodegradable films	Kefiran plasticized with glycerol,d-glucitol,d-galactitol,d-mannitol, andd-limonene	Stable, flexible films without porosity or cracks	Food packaging	[119]
LAB from kefir grains	Kefiran	Edible composite films/emulsified films	Kefiran emulsified with oleic acid (OA)	Reduced water vapor permeability of the emulsified films	Films for some food applications that require a low affinity toward water	[176]
*Leuconostoc pseudomesenteroides* R2	Dextran	Natural film	Dextran	Thermal stability, resistance to sterilization, pseudoplastic behavior, stability in acidic and alkaline conditions	Food packaging	[103]
*Lactobacillus plantarum* strains (*L. plantarum* LP3, *L. plantarum* AF1, and *L. plantarum* LU5)	-^1^	Bioactive edible film	Konjac glucomannan and probiotic *Lactobacillus plantarum* strains	Preserving the color and the ascorbic acid from the fresh-cut kiwifruit.	Extended shelf life of Kiwi	[177]
*Liquorilactobacillus (L.)* sp. *CUPV281* and *Liquorilactobacillus (L.) maliCUPV271*	Dextran	Films obtained by casting and compression	Soy protein with exopolysaccharides	Transparent and homogeneous resistant to temperatures up to 190 °C, with an inhibitory effect on the germination of fungal spores	Films for food applications	[178]
Lactic acid bacteria	Dextran	Composite Films	Chitosan-based composite films blended with dextran, plasticized with 1,3-propanediol	Thermally and mechanically resistant, antioxidant properties and biodegradablefeatures	Food packaging	[122]

-^1^ Not specified.

## Data Availability

The data presented in this study are available within the article.

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
