# Peer review of "Biopolymers Produced by Lactic Acid Bacteria: Characterization and Food Application"

_polymers, 2023, doi:10.3390/polym15061539_

Round 1
Reviewer 1 Report
The review describes in great detail the structure of polymers produced by lactic acid bacteria, as well as concisely and competently describes the modern classification of lactic acid bacteria. I consider the positive side of the review to be the description of the main pathways of carbohydrate metabolism by lactic acid bacteria (the homoenzymatic pathway and the heteroenzymatic pathway), which have been studied in sufficient detail, have long been known and are described in this review taking into account modern positions. The review is necessary for readers, since the relevance of this topic is growing in connection with new environmental requirements. The volume of the review is impressive, there are shortcomings and typos, so below are recommendations for authors to quickly improve the text.
Recommendations:
1) According to the text of the article, instead of the “degree” sign (º), there is the number 0, which somewhat complicates the perception of the text and confuses.
2) Lines 314-316 include this statement: "The use of thermotolerant strains (such as L. rhamnosus that can grow at 42°C) can minimize contamination problems in lactic acid production." I would like a more detailed presentation of the question of how the thermotolerance of bacteria can contribute to the reduction of contamination in the production of lactic acid.
3) When the conditions for the biosynthesis of exopolysaccharides by lactic acid bacteria are discussed (lines 328-335), it is indicated that manipulations with cultivation parameters lead to an increase in the yield of exopolysaccharide, but it is not specified how exactly the cultivation conditions should be changed and how much the yield can be increased. Also, when discussing the genetically engineered change in the Pediococcus acidilactici strain (lines 426-430), it is not indicated how much the output of D-lactic acid will increase as a result of this intervention.
4) The order of presentation of the material for each genus of lactic acid bacteria in section 3.2 is somewhat unclear: first lactobacilli are described, then lactococci, then leuconostocs, streptococci, enterococci and again leuconostocs - what is the purpose of breaking the information about leuconostocs? Do the authors have a special vision to structure the presentation of the material in this way? Similarly, information is duplicated almost verbatim about polysaccharides capable of forming films: lines 936-941 and lines 948-952.
5) In general, some negligence of presentation can be noted - in particular, well-known information is duplicated several times in the text, for example, about the pathways of metabolism of carbohydrates by lactic acid bacteria, it is also deciphered several times in the text what homo- and heteropolysaccharides are, and what monomers are included compound.
6) I really liked Table 1 - Selected studies of biopolymers produced by lactic acid bacteria. The table shows the names of the producers, the initial carbohydrate substrate, which polymer is synthesized as a result, the cultivation conditions and the yield of biopolymers are specified.
7) It should also be noted the information given in the article about the positive experience of using consortiums of microorganisms for the production of biopolymers - they allow the use of industrial wastes containing various carbon substrates as nutrient media.
8) Typos and shortcomings:
Lines 74-85. Links to an electronic resource should be included in the list of references properly or discarded,
You need to work more carefully with tables,
Check if links are formatted correctly
Author Response
Respected reviewer,
Please see the attachment.
Thank You very much!

Reviewer 2 Report
This article reports a review regarding the comprehensive information on the numerous biopolymers biosynthesized by lactic acid bacteria (LAB) and their applications in food packaging. Overall, this work is well written. Some strengths are as follows: This topic is interesting and significant in the field of biodegradable packaging. The topic is original. This review includes updated information regarding the recent literature. The conclusions are consistent with the evidence and arguments and address the main point of the review. The references are appropriate. Only a few minor points are required to be improved.
1) The authors are invited to write more about the LAB taxonomy. The taxonomy tree is recommended to be provided as a Figure to make it easier to understand.
2) For section 4, which describes the biosynthesis of different biopolymers, the chemical equations and/or reaction pathways are recommended to be added. There are only texts in this present manuscript.
3) Please recheck typos and formatting errors. There are some mistakes, such as:
- Use of regular font instead of italics for the scientific names of bacteria
- At line 229, please add the symbol ° to “200C and 450C”.
- At line 233, what is meant by “2,3-bu109ediol”?
- There are some more errors. Please carefully check.
Author Response
Respected Reviewer,
Please see the attachment including the point-by-point responses to Your comments.
Thank You very much!
